# Hyaluronic Acid-Targeted Stimuli-Sensitive Nanomicelles Co-Encapsulating Paclitaxel and Ritonavir to Overcome Multi-Drug Resistance in Metastatic Breast Cancer and Triple-Negative Breast Cancer Cells

**DOI:** 10.3390/ijms22031257

**Published:** 2021-01-27

**Authors:** Vrinda Gote, Amar Deep Sharma, Dhananjay Pal

**Affiliations:** Division of Pharmacology and Pharmaceutical Sciences, School of Pharmacy, University of Missouri-Kansas City, 2464 Charlotte Street, Kansas City, MO 64108, USA; vrindagote@mail.umkc.edu (V.G.); ashw6@mail.umkc.edu (A.D.S.)

**Keywords:** CD44 receptors, MCF-7, MDA-MB-231, MCF-12A, disulfide bond, glutathione, mitochondrial membrane potential, VEGF, cellular uptake and intracellular distribution, design of experiment, stability studies, active targeting

## Abstract

Active targeting and overcoming multi-drug resistance (MDR) can be some of the important attributes of targeted therapy for metastatic breast cancer (MBC) and triple-negative breast cancer (TNBC) treatment. In this study, we constructed a hyaluronic acid (HA)-decorated mixed nanomicelles-encapsulating chemotherapeutic agent paclitaxel (PTX) and P-glycoprotein inhibitor ritonavir (RTV). HA was conjugated to poly (lactide) co-(glycolide) (PLGA) polymer by disulfide bonds (HA-ss-PLGA). HA is a natural ligand for CD44 receptors overexpressed in breast cancer cells. Disulfide bonds undergo rapid reduction in the presence of glutathione, present in breast cancer cells. The addition of RTV can inhibit the P-gp and CYP3A4-mediated metabolism of PTX, thus aiding in reversing MDR and sensitizing the cells toward PTX. An in vitro uptake and cytotoxicity study in MBC MCF-7 and TNBC MDA-MB-231 cell lines demonstrated the effective uptake of the nanomicelles and drug PTX compared to non-neoplastic breast epithelium MCF-12A cells. Interestingly, in vitro potency determination showed a reduction in mitochondrial membrane potential and reactive oxygen species in breast cancer cell lines, indicating effective apoptosis of cancer cells. Thus, stimuli-sensitive nanomicelles along with HA targeting and RTV addition can effectively serve as a chemotherapeutic drug delivery agent for MBC and TNBC.

## 1. Introduction

Cancer is the leading causes of morbidity and mortality worldwide. Breast cancer is the most frequently diagnosed cancer among US women and is second leading cause of cancer death [1]. More than 90% of breast cancer-related mortality are associated with tumor metastasis, also called stage IV breast cancer [2]. Heterogeneous breast cancer cells can gain mortality by the activation of the epithelial-to-mesenchymal transition (EMT) pathway. EMT is a major contributor of the metastasis of epithelial-originated breast cancer. Triple-negative breast cancer (TNBC) is an aggressive metastatic breast cancer (MBC) cancer characterized by the absence of estrogen receptor (ER), progesterone receptor (PR), and human epidermal growth factor receptor 2 (HER2) [2]. TNBCs can have a high rate of recurrences (<5 years) and systematic metastases [3]. MBC and can be less sensitive to current treatments, like chemotherapy, due to the lack of targeted therapies and occurrence of multi-drug resistance (MDR) [4,5,6].

For several decades, conventional cytotoxic agents have represented the foundation of anticancer chemotherapy. Most of these cytotoxic agents exert their antitumor effects through interaction with DNA or its precursors or microtubule polymerization, thereby killing or preventing further growth of the rapidly dividing tumor cells. Cancer chemotherapy represents the most common treatment modalities for the disease. However, a major impediment to successful chemotherapy is the development of MDR [7,8].

From a pharmaceutical point of view, drug bioavailability at therapeutic concentrations in the target cancer cells is the key to success of chemotherapy. When cells develop drug resistance, chemotherapeutic agents are accumulated less in cancer cells due to higher efflux and faster metabolism by overproducing metabolizing enzymes cytochrome P450 (CYP3A4). Therefore, less drug absorption along with higher metabolism leads to a sub-therapeutic concentration at the target site, resulting in treatment failure. We have previously examined a strategy of combining ritonavir with anticancer agents to control drug efflux, metabolism and allow sufficient drug entry into tumor cells in vitro [9].

Vadlapatla et. al. demonstrated that in response to vinblastine, human colon adenocarcinoma cells (LS-180) overexpressed multi drug resistance protein 1 (MDR1), MRP2, CYP3A4, and pregnane X receptor (PXR) within 72 h [1]. When co-administered anti-cancer agents with ritonavir (RTV), overexpression of these genes could be suppressed or reduced. These results suggest an enhanced activity of chemotherapeutics in the presence of RTV. In summary, the combination therapy of anticancer drugs with RTV can overcome drug resistance by deactivating the overexpression of efflux transporters and metabolizing enzymes. In another study, we demonstrated the inhibition of the hypoxia-induced overexpression of HIF-α and, consequently, overproduction of VEGF and endothelial proliferation by [10]. Therefore, we anticipate that anti-cancer drug regimens containing RTV would enhance the therapeutic exposure of cancer cells to anticancer agents, potentially improving chemotherapeutic efficacy, devoid of drug resistance.

The next problem of chemotherapy is high toxicity. Most of the chemotherapy agents deliver very high doses without any targeting moieties and, consequently, drugs distribute to every cell. Cancer cells receive a sub-therapeutic concentration of chemotherapeutic drugs due to nontargeted therapy and MDR. Therefore, it appears that an appropriate drug delivery strategy can acquire successful chemotherapy and eradicate cancer. This highlights the urgent need to develop effective, targeted therapies for MBC.

Self-assembling nanomicelles encapsulating hydrophobic cancer drugs have emerged as a preferred drug carrier for various types of cancer [11,12,13]. These nanomicelles can have numerous advantages such as (i) increasing the water and serum solubility of hydrophobic drugs, (ii) improve circulation time of the drug in the serum, (iii) improve tumor penetration and accumulation of the drugs by enhanced retention and the permeation effect and (iv) controlled and sustained release of the drug at the site of action [14,15,16]. Chemotherapeutic drugs encapsulated in nontargeted conventional nanomicelles and nanoparticles show a controlled release behavior, sometimes for weeks when the nanocarrier is deposited at the site action. This may lead to MDR in the cancer cells [13,17,18].

Stimuli-responsive nanocarriers can achieve effective release of chemotherapeutic drugs in the tumor tissue. Such nanocarriers can release their cargo in response to various extracellular and intracellular biological stimuli (e.g., acidic pH, upregulated enzymes, altered redox potential), along with external artificial stimuli like magnetic fields, light, and temperature. Such nanocarriers are called “smart” nanocarriers for the delivery of anticancer drugs [12,19,20,21,22,23]. Reduction-sensitive nanocarriers, a type of smart nanocarrier, are composed of polymers having disulfide bonds [24]. Disulfide bonds rapidly degrade in a cellular environment with a higher presence of glutathione (GSH), commonly found in cancer cells. However, such bonds are highly stable in the extracellular fluid and systemic circulation with lower amounts of GSH [13]. GSH levels are approximately four times higher in breast, ovarian, head and neck, and lung cancer as compared to the normal cells [25,26,27]. Breast cancer cell lines like MCF-7 and MDA-MB-231 also express substantially higher levels of GSH, which can be exploited for disassembly of smart, reduction-sensitive nanomicelles [28,29].

Paclitaxel (PTX) is a potent microtubule-stabilizing chemotherapeutic drug approved by the US-FDA for use in breast cancer. The drug arrests the cell cycle at the mitotic stage, causing death of the cancerous cells [30]. PTX loaded in nanocarriers is increasingly being investigated for a variety of cancers like breast, ovarian, Kaposi’s sarcoma, lung, head and neck cancer, and prostate and cervical cancer [31]. Although PTX is extremely useful in killing cancer cells, the drug can cause side effects like nephrotoxicity and neurotoxicity and kill normal breast cells. Higher doses may lead to a higher systemic toxicity associated with MDR. Breast cancer cells overexpress multidrug-resistant proteins (MRPs) like P-glycoprotein (P-gp) and breast cancer-resistant protein (BCRP). These pumps efflux anticancer drugs out of the cancer cells, causing MDR [28,32,33,34,35,36]. RTV is a P-gp inhibitor and acts by blocking the efflux pump and increasing the intracellular concentration of P-gp substrates [37,38,39,40,41]. The drug also inhibits metabolic enzyme CYP3A4, which reduces the metabolism of taxenes like PTX and increases the bioavailability and intracellular concentration of PTX [38].

Chemotherapeutic agents targeted toward the cancer cells can greatly help in reducing cytotoxic effects of the treatment on healthy cells. This can be achieved by conjugating the active targeting moiety to nanocarriers for selective accumulation in cancer cells. Hyaluronic acid (HA) is a naturally occurring ligand composed of repeated units of disaccharides like b-1,4-D-glucuronic acid–b-1,3-N-acetyl-D-glucosamine. HA is a glycosaminoglycan and a natural component of the vitreous humor. HA is a biocompatible, biodegradable, and nonimmunogenic biopolymer capable of actively targeting Cluster of differentiation-44 (CD44) on cell surface receptors overexpressed in many cancer cells [42]. CD44 receptors are responsible for cell adhesion, migration, orientation, and chemoresistance [43]. CD44 receptors are overexpressed on breast cancer cells like MCF-7 and TNBC cells like MDA-MB-231. In addition, HA is ligand for lymphatic vessel endothelial hyaluronan receptor 1 (LYVE1) and hyaluronan-mediated motility receptor (RHAMM) [43]. The affinity of chemotherapeutic nanocarriers for cancer cells can be enhanced by HA targeting. This can allow the receptor-mediated endocytosis pathway in cancer cells.

In this study, we designed a bio-inspired “smart” nanomicellar drug delivery system encapsulating PTX and RTV for MBC. The nanomicelles were composed of two polymers, Vitamin E TPGS (Vit E TPGS) and hyaluronic acid-targeted poly (lactide) coglycolide (HA-ss-PLGA). Vit E-TPGS is an amphiphilic polymer that can self-assemble to form micelles. The polymer also functions by inhibiting P-gp and helps by reversing MDR in cancer cells. HA-ss-PLGA was used as a diblock polymer having HA as the hydrophilic part and PLGA as the hydrophobic part. This polymer can provide structural stability to the nanomicelles. HA was linked to PLGA via a disulfide bond, which could be cleaved by GSH present in the cytosol of cancer cells, releasing PTX and RTV. HA was used as a targeting agent that can actively accumulate within breast cancer cells via CD44 receptor-mediated endocytosis. PTX can selectively accumulate within the breast cancer cells due to P-gp inhibition primarily by RTV. This smart nanomicellar formulation was designed to selectively target breast cancer cells and reverse MDR.

## 2. Results and Discussion

### 2.1. Synthesis and Characterization of HA-ss-PLGA Copolymer

In this study, active targeting was achieved by creating a graft PLGA polymer and conjugating it with HA. The targeting agent: HA can preferably reach breast cancer cells due to their high expression of the CD44 receptor. In addition to this strategy, HA was conjugated to PLGA via a disulfide (s–s) bond. This bond in the drug delivery system can undergo reduction and be cleaved at the site of action to release the active drug. Drug delivery systems and antibody drug conjugates containing disulfide bonds can be a very effective anti-cancer therapeutic [6,7]. The complete synthesis scheme for HA-ss-PLGA graft co-polymer is depicted in Figure 1. PLGA polymer was first synthesized by ring-opening condensation of L-lactide and glycolide polymers. The functionalized PLGA (PLGA-NHS) was reacted with the amine-functionalized HA (HA-CYS) for the formation of HA-ss-PLGA. Cystamine dihydrochloride (CYS) was selected as a linker due to the presence of the disulfide bond.

The successful formation of HA-ss-PLGA was confirmed by H^1^ NMR by comparing it to the HA and PLGA H^1^ NMR spectrum, as seen in Figure 2. It is an overlay of the H^1^ NMR spectrum corresponding to (A) HA-ss-PLGA, (B) PLGA, and (C) HA. In Figure 2C, peaks at δ 3.17–4.12 ppm correspond to glycoside protons of the HA polysaccharide chain. The peak at δ 1.86 corresponds to the characteristic peak of the methyl group (-NHCOCH_3_-) in HA. In Figure 2B, peaks at δ 5.2 and δ 4.8 ppm are related to methine (-COCH(CH_3_)-) and methyl protons (-COCH(CH_3_) O-) of the PLGA segment of the copolymers. The peak at δ 1.96 ppm corresponds to the methyl group (-CH_3_-) protons of the lactide part of PLGA. In Figure 2A, peaks at δ 1.79 correspond to the methyl group (-CH_3_-) protons of the lactide part of PLGA. It is shifted toward the right as compared to the PLGA spectrum. Relatively less intense peaks at δ 2.47 ppm are the special peaks of methylene of CYS. The peaks at δ 3.02 to 3.64 correspond to the characteristic peaks due to the hydroxyl (-OH) group present at various benzene rings in HA polysaccharide. Thus, the peaks of HA and PLGA are present in the H^1^ NMR spectrum for HA-ss-PLGA, Figure 2A. This confirms the synthesis of HA-ss-PLGA graft co-polymer.

### 2.2. Design of Experiment and Formulation Optimization

Formulation parameters of hyaluronic acid targeted paclitaxel and ritonavir nanomicellar formulation (HA − PTX + RTV − NMF) were designed and optimized using a full factorial design of experiment (DOE) in JMP^®^ software. This enabled the monitoring of effects of the independent variables on the dependent variables. The independent variables were selected as X1: sonication time (min), X2: HA-PLGA (wt%), and X3: Vit E-TPGS (wt%), while the dependent variables were selected as Y1: size (nm), Y2: polydispersity index (PDI), Y3: zeta potential (mV), Y4: PTX EE (entrapment efficiency) (%), Y5: PTX LE (loading efficiency) (%), Y6: RTV EE (%), and Y7—RTV LE (%). The full factorial DOE generated 11 combinations or runs of independent variables for analyzing their effect on dependent variables. A higher (1), middle (0), and lower (−1) value was selected for each independent variable. The initial combination of independent variables is represented in Table 1.

The final output of all the DOE is summarized in Table 2. HA − PTX + RTV − NMF (F1 to F11) were prepared by the solvent evaporation and film rehydration methods. Results from analyzing nanomicellar formulations F1-F11 were added to the full factorial DOE and a least squares analysis of the independent variables on the dependent variables was performed (Table 2). For each dependent variable, a Pareto chart, actual-by-Predicted plot, surface profiler, final prediction profiler, and equations were generated (Figure 3). The prediction equations for dependent variables are as follows:Y1 = 1005.4 − 53.6 × X1 + 266.4 × X2 + 185.8 × X3 × X1 − (X2 × 11.1) × X1 − (X3 × 7.9) × X2 − (X3 × 8.3)Y2 = 0.23 + 0.02 × x1 + 0.06 × X2 + 0.05 × X3 × X1 + (X2 × 0.01) × X1 + (X3 × 0.002) × X2 + (X3 × 0.06)Y3 = 0.2 +0.04 × X1 − 0.6 × X2 + 0.5 × X3 × X1 − (X2 × 0.3) × X2 + (X3 × 0.05)Y4 = 184 − 5.2 × X1 − 22.1 × X2 − 26.2 × X3 × X1 + (X2 × 0.74) × X1 + (X3 × 1.34) × X2 + (X3 × 1.83)Y5 = 57.3 − 2.2 × X1 − 801 × X2 − 9.8 × X3 × X1 + (X2 × 0.4) × X2 + (X3 × 0.5)Y6 = 355 − 11.3 × X1 − 57.2 × X2 − 52.8 × X3 × X1 + (X2 × 1.8) × X1 + (X! × 2.2) × X2 + (X3 × 4.6)Y7 = 20.3 − 0.74 × X1 − 3.23 × X2 − 3.11 × X3 × X1 + (X3 × 0.12) × X2 + (X3 × 0.21)where Y1—size (nm), Y2—PDI, Y3—zeta potential (mV), Y4—PTX EE (%), Y5—PTX LE (%), Y6—RTV EE (%), Y7—RTV LE (%), X1—sonication time (min), X2—HA-PLGA (wt%), and X3—Vit E-TPGS (wt%)

The Actual-by-Predicted plots provide a visual representation of how well the model fits and also compares the variation occurring due to the dependent variables [8]. It also gives a correlation between the results obtained from experimentations and the outcomes predicted by the software. Actual-by-Predicted plot for size, PDI, zeta potential, PTX-EE, PTX-LE, RTV-EE, and RTV-LE have probability (*p*) values of 0.87, 0.86, 0.91, 0.85, 0.14, 0.38, and 0.048, respectively, Figure 3A(i)–G(i). This implies a positive correlation of dependent variables (size, PDI, zeta potential, PTX-EE, and RTV-EE) with independent variables (sonication time, HA-PLGA, and Vit-E-TPGS), while dependent variables PTX LE and RTV LE do not show a high positive or negative correlation with the independent variables. This can imply that these factors are not affected much by the change in the independent variables.

In addition, contour plots for size, PDI, zeta potential, PTX-EE, PTX-LE, RTV-EE, and RTV-LE, as seen in Figure 3A(ii)-G(ii), were developed. These are three-dimensional representations of changes in dependent variables on shifting independent variables. When the independent input variables were set to X1: sonication time (22.5 min), X2: HA-PLGA (2 wt%), and X3: Vit E-TPGS (4 wt%), a lower size, PDI, and zeta potential and higher PTX-EE, PTX-LE, RTV-EE, and RTV-LE were predicted. Along with the above analysis of the relationship between independent and dependent variables, a prediction profiler for the variables was established. The prediction profiler could help in adjusting the levels of independent variables in specific combinations, where the values of dependent variables can be predicted. The results obtained from the DOE model were comparable to the results obtained by performing the actual experiments. This can indicate the robustness of the model in predicting the desired outcomes.

Before developing the prediction profiler (Figure 4), independent factors like size, PDI, and zeta potential were set to minimum desirability and loading, and entrapment efficiencies were set to maximum desirability. Input variables sonication time (22.5 min), HA-PLGA (2 wt%), and Vit E-TPGS (4 wt%) resulted in lower nanomicellar size (144.0 nm), lower PDI (0.2), lower zeta potential (0.2), and higher PTX-EE (96.5%), PTX-LE (7.4), RTV-EE (95.5%), and RTV-LE (1.9), as seen by the experimental data (Table 2). Independent variables sonication time, HA−ss−PLGA, and Vit E-TPGS were set to the initial coded design F1 = (0)(0)(0) (Table 2) in the prediction profiler, and the desirability value of the predication profiler was noted. The desirability value of the prediction profiler (Figure 4) DOE full factorial model was 0.97. A desirability value of 1.0 indicated the highest correlation between the prediction mathematical model and experimentally obtained data. The simulations obtained from the prediction profiler were in good agreement with the results acquired by actual experimentation specifically for Formulation-1 (F-1). Hence, the specific combination of independent variables in F1 in Table 2 was used for further experimentation. Experiment simulation results by JMP 10.0 software and results obtained from the actual experiments suggest that polymer concentration and sonication time when kept at certain levels can yield a HA − PTX + RTV − NMF with low size, PDI, and zeta potential and high loading and high entrapment efficiencies. Hence, F-1 showing optimization of all the dependent outcomes in the simulation studies and in the actual experiment performed was selected as the optimized formulation for further studies.

### 2.3. Formulation Characterization

#### 2.3.1. Size, PDI, and Zeta Potential

Amphiphilic polymers in aqueous solution at specific conditions form spherical nanomicelles. These have a hydrophobic core and hydrophilic corona facing the exterior aqueous environment. Such nanomicelles can be utilized to entrap hydrophobic drugs in their core. Thus, they can serve as excellent drug delivery vehicles for hydrophobic drugs and support in improving their solubility [9,10,11]. Here, we prepared HA − PTX + RTV − NMF using a mixture of polymers like HA−ss−PLGA and Vit E-TPGS. The targeted nanomicelles were prepared by the solvent evaporation film rehydration method. The solvent and drug film were rehydrated using HPLC-grade water. HA − PTX + RTV − NMF -prepared F1-F11 formulations in Table 2 had a hydrodynamic size range of 142.5–256 nm, PDI of 0.2–0.8, and zeta potential of 0.01–0.023 mV. Nanomicelles with smaller size (<300 nm) and neutral to positive charge can allow better absorption into cells and allow better tissue penetration. As seen in Table 2, in full factorial DOE, when the independent variables like sonication time of HA−ss−PLGA and Vit E-TPGS are present in their middle values of 22.5 min, 2 wt% to 4 wt%, respectively, a formulation with minimum size, near zero zeta potential, and minimum PDI can be obtained. As seen in Table 2, increasing the concentration of H−ss−PLGA can increase the size of the nanomicelles, which is not entirely desirable. The PDI of such formulations was high. Alternatively, increasing the concentration of Vit E-TPGS also increased the size, but to a lesser extent than the size increase by HA−ss−PLGA. An excessive use of Vit E-TPGS can actually weaken the interactions between hydrophilic fragments of the polymer constituting the nanomicelles [12]. The zeta potentials of all the nanomicelles were between 0.1 and 1.0 mV. The ratio of HA−ss−PLGA: Vit E-TPGS 2:4 along with 22.5 min allows one to obtain the optimized formulation because it resulted in nanomicelles with almost zero zeta potential. Transmission electron microscopy (TEM) image of HA − PTX + RTV − NMF depicted spherical nanomicelles. Figure 5 depicts the TEM image (A), size distribution by intensity (nm) (B) and apparent zeta potential (mV) for HA − PTX + RTV − NMF.

#### 2.3.2. Entrapment and Loading Efficiencies

PTX and RTV entrapped in HA − PTX + RTV − NMF were measured by the direct quantification method using reverse micellization. For this process, the nanomicelles were freeze dried, and dichloromethane (DCM), was added. This allowed the inner hydrophobic drug to be exposed to the surrounding organic solvent. This was followed by the evaporating DCM to obtain a thin polymer and drug film. This film was dissolved in the mobile phase. This ensured dissolution of the hydrophobic drugs in the nanomicelles in the mobile phase. The final samples were centrifuged before analysis. The drug release was quantified by UHPLC using the method described in the methods section. Depending on the variations in the total amphiphilic polymer (HA−ss−PLGA and Vit E-TPGS), the EE varied from 72.71 to 97.13% and the LE varied from 0.55 to 10.9% in F1–F11 (Table 2). Among formulations F1–F11, F-1, F-2, and F-5 exhibited the lowest size, lowest zeta potential, and highest PTX and RTV EE. F-7 and F-8 had the highest LE for PTX and RTV but the highest nanomicellar size, which was not desirable. In addition, the LE of the drugs in F-7 and F-8 can be explained by the lower polymer content in these formulations as compared to the others. Nanomicellar formulations that had higher polymer contents like F-4 and F-9 did not have the highest EE or lowest nanomicellar size. Nanomicellar formulations F-1, F-2, and F-5 have the same composition. The dependent values of variables for F-1, F-2, and F-5 represent the middle points for the DOE. Here, we decided to triplicate the middle point formulation in the DOE to see the robustness of the DOE and its reproducibility in the results. It was observed that formulations F-1, F-2, and F-5 having independent variables set to sonication time: 22.5 min, HA−ss−PLGA 2.0 wt%, and Vit E-TPGS 4.0 wt% demonstrate optimized desired dependent variables. Hence, the nanomicellar formulation F-1 (the same as F-2 and F-5) was selected as the optimized formulation and was selected for all further experiments.

### 2.4. Critical Micellar Concentration

Critical micellar concentration (CMC) is an important determinant, directing the performance of the drug delivery system. The efficient delivery of the drug to its site of action with minimal degradation in the circulation is determined by the CMC of the polymers used to construct the nanomicelles. CMC can ultimately affect the drug bioavailability [13]. A higher value of CMC for a given polymer can result in the premature release of the drug in the systemic circulation. Enzymes and proteins in the serum can result in destruction of the nanomicellar structure. This can cause premature drug release at a nontarget area and loss of functionality [14]. This loss can be ameliorated by encapsulating hydrophobic drugs in the core of nanomicelles with lower CMC [15]. A lower CMC value for a nanomicellar formulation helps in the prevention of disruption of the nanomicellar structure after dilution into the systemic circulation [16,17]. In this study, CMC was determined for HA−ss−PLGA, Vit E-TPGS, and a mixture of HA−ss−PLGA and Vit E-TPGS in the F-1-optimized formulation ratio: 2.0:4.0 wt%. The purpose of this study was not only to calculate the CMC of the polymers but also to determine whether the mixture of polymers had any effect on the total CMC value. The CMC of HA−ss−PLGA, Vit E-TPGS, and a mixture of HA−ss−PLGA and Vit E-TPGS was 0.86 wt%, 0.58 wt%, and 0.52 wt%, respectively. A lower CMC value of the mixture of the polymers indicated that the polymeric mixture of HA-ss-PLGA and Vit E-TPGS could form stable nanomicelles than the individual polymers.

### 2.5. Effect of Dilution and Temperature on Storage Stability

Nanomicellar formulations when administered into the systemic circulation undergo rapid dilution. It is important for the formulation to stay intact and deliver the cargo drug at the site of action. The site of action, in this case, the MBC cells, is not the same as the site of administration. Thus, the nanomicelles must travel a substantial distance ensuring the polymer and shape of nanomicelles remain intact. Nanomicellar integrity can be experimentally determined by the dilution study. Here, HA − PTX + RTV − NMF was diluted up to 200 times using HPLC-grade water. Diluted nanomicelles were stored for 24 h at three temperatures, 4 °C, room temperature (RT) which is approximately 22 °C, and 37 °C. This was followed by analyzing the critical quality attributes (CQAs) of the nanomicelles like size, PDI, and zeta potential. One milliliter of the resultant diluted HA − PTX + RTV − NMF was analyzed for the change in nanomicellar size characters. Table 3 depicts the effect of dilution on size, PDI, and zeta potential. As the F1 formulation showed the lowest CMC (0.52 wt%), this F1-optimized formulation was used for this dilution and other studies.

The effect of dilution on nanomicellar storage stability at various temperatures indicates that dilution has the least effect on the CQAs of HA − PTX + RTV − NMF diluted up to 100 times when stored at 4 °C. The same is also true for the CQAs of HA − PTX+RTVNMF diluted up to 100 times stored at 22 °C for 24 h. A higher level of dilution causes slight changes in CQAs, while HA − PTX + RTV − NMF dilution resulted in an increased size of the nanomicelles by 20 nm, PDI by 0.4, and zeta potential by 1.5 mV when stored at body temperature for 24 h. Storage at higher temperature results in destabilization of the polymers to a certain degree. This allows access of the water molecules in the nanomicellar core and corona, resulting in increased size and zeta potential.

### 2.6. Temperature Stability Studies

Storage stability and plasma stability studies for HA − PTX + RTV − NMF were conducted by analyzing the CQAs of the formulation of size, PDI, and zeta potential. Storage stability involved the storage of HA − PTX + RTV − NMF at temperatures of 4 °C, 25 °C, and 40 °C. Regular samples were withdrawn at Time 0, Day 3, and Day 7 (Table 4). There was no significant difference in the CQAs of HA − PTX + RTV − NMF when stored at 4 °C and 25 °C, up to 7 days. However, when the same formulation was stored at 40 °C, there was a significant increase in size, zeta potential, and PDI of the formulation. For plasma stability studies, the above nanomicellar formulations suspended in HPLC-grade water were stored in 10% FBS in Dulbecco’s phosphate-buffered saline (DPBS). The formulations were stored at body temperature, which is 37 °C in a water bath with constant shaking to simulate physiological conditions. Here, 1.0 mL of the sample was withdrawn at Day 3 and Day 7. The CQAs were compared to that at Time 0. The results are presented in Table 5. There was a significant increase in size (1.7 times) up to Day 3 and at Day 7 for the HA − PTX + RTV − NMF. This suggested that the nanomicellar formulation can be stable in the circulation up to three days.

### 2.7. Dissolution and Drug Release

In vitro dissolution and drug release for PTX from HA − PTX + RTV − NMF and PTX + RTV − NMF were determined by the UHPLC-MS method mentioned in the methods section. Briefly, 1.0 mL of HA − PTX + RTV − NMF was transferred to a dialysis tubing of a cut-off molecular weight of 2000 KDa. The dialysis tubing was surrounded externally by a 5 mL buffer solution of (i) PBS at pH 7.4, (ii) phosphate buffer saline (PBS) at pH 6.8, (iii) DPBS with 10% fetal bovine serum (FBS) at pH 7.4, and (iv) DPBS with 10% FBS at pH 6.8 in a 15 mL centrifuge tube. Samples of the external fluid were collected at predetermined time points and the PTX release was determined. The concentration of PTX in various buffer solutions is depicted in Figure 6. The cumulative PTX release in Figure 6A was greater at a lower pH of 6.8 as compared to physiological pH 7.4. The PTX release from the HA − PTX + RTV − NMF and PTX + RTV − NMF in PBS at pH 7.4 was 43.5 and 50%, respectively. Similarly, the PTX release from the HA − PTX + RTV − NMF and PTX + RTV − NMF in PBS at pH 6.8 was 76.8 and 65.5%, respectively. The targeted and nontargeted nano formulations were also analyzed for PTX release in DPBS buffer containing 10% FBS. Such buffer media can simulate serum protein conditions and can be a better tool for in vitro–iv vivo correlation (IVIC). The prepared DPBS buffer with 10% FBS was made into two buffer solutions with different pH values (7.4 and 6.8). DPBS buffer with 10% FBS at pH 6.8 can simulate conditions similar to the tumor site. Figure 6B depicts the cumulative PTX release in two buffer solutions, all containing 10% FBS in DPBS buffer. The PTX release from HA − PTX + RTV − NMF and PTX + RTV − NMF in 10% FBS at pH 7.4 was 74.5 and 67.3%, respectively. Similarly, the PTX release from HA − PTX + RTV − NMF and PTX + RTV − NMF in PBS at pH 6.8 was 98.2 and 93.2%, respectively. These results indicate that an almost complete release of PTX from the two nano-formulations was achieved in 10% FBS at pH 6.8.

### 2.8. Nanomicelles in Reduction Stimulated Environment

GSH is expressed in higher quantities in the cytoplasm of cancer cells than normal cells [18,19,20,21]. GSH acts on disulfide bonds and cleaves them to the thiol bond. It can trigger the disassembly of the nanomicelles and accelerate drug release from their core. Disulfide bonds are fairly stable in the extracellular environment, but can rapidly disintegrate in a reductive intracellular environment, enriched with GSH, through a thiol-disulfide exchange reaction [22]. To demonstrate if the presence of GSH can trigger the reduction of HA from PLGA polymer in the nanomicellar formulation, HA − PTX + RTV − NMF and PTX + RTV − NMF were incubated in 50 mM GSH solution for 48 h in a water bath at 37 °C. The nanomicelles were analyzed for their size at various time points. As a control, each formulation was placed in a solution with 0 mM GSH. As seen in Figure 7A, the size of HA − PTX + RTV − NMF in GSH increased from 152.3 ± 2.5 nm at 0 time points to 1354 ± 45.9 nm at 48 h. There were approximately nine time increases in size of the HA-targeted formulation. The change in size of the same nano-formulation without GSH was negligible (195.6 ± 7.5 nm at 0 h to 300.3 ± 19.5 nm at 48 h). As expected, for the nanoformulations without the disulfide bond and external GSH, the change in size was negligible. HA − PTX + RTV − NMF in the presence of GSH also resulted in larger aggregates. Thus, it can be predicted that nanomicelles having a disulfide bond like HA − PTX + RTV − NMF can disassemble in the presence of GSH.

To further evaluate if HA − PTX + RTV − NMF can release the drug in a reductive environment of GSH, cumulative PTX release from the above experiment was evaluated. It is essential for a drug delivery system to effectively deliver the payload at the site of action, followed by an effective amount of drug release for the desired pharmacological activity. Anticancer drug delivery vehicles, specifically stimuli-sensitive nanocarriers, should result in effective drug release when the desired stimulus environment is applied for appropriate anticancer activity. Here, HA − PTX + RTV − NMF disassembly takes place in the presence of external GSH. The amount of PTX released from the formulations was determined by the UHPLC-MS method. Briefly, HA − PTX + RTV − NMF and PTX + RTV − NMF were incubated in 50 mM GSH in PBS solution. As a comparison, the formulations were also incubated in PBS without GSH, as stated above. The same samples used for size determination were applied to calculate cumulative PTX release. The samples were filtered, evaporated, dissolved in the mobile phase, and centrifuged for cumulative PTX determination. As seen in Figure 7B, the cumulative PTX release was maximum for HA − PTX + RTV − NMF incubated in GSH. This also correlates with the increase in size of the formulation. This indicates that GSH not only causes the disassembly of nanomicelles with disulfide linkage, but also results in an almost complete drug release from the formulation. In the same figure, in the absence of GSH, approximately 20–30% of PTX was released from all the formulations. These results also coincide with the drug release behavior of the HA-targeted and HA-nontargeted nanomicelles in PBS at pH 7.4.

### 2.9. Cellular Uptake and Intracellular Distribution Study

#### 2.9.1. Cellular Uptake by fluorescence-assisted cell sorting (FACS)

The in vitro cellular uptake of FITC-labeled HA − PTX + RTV − NMF and FITC-labeled PTX + RTV − NMF was determined in MCF-7, MDA-MB-231, and normal breast epithelium MCF-12A cells to quantitate the uptake and the nanomicellar formulation. This study also helped to compare the uptake of the targeted nanomicelles (HA − PTX + RTV − NMF) as compared the nontargeted (PTX + RTV − NMF) nanomicelles. HA − PTX + RTV − NMF and PTX + RTV − NMF were labeled with FITC, as mentioned above. MCF-7, MDA-MB-231, and MCF-12A cells were treated with FITC-labeled HA − PTX + RTV − NMF or FITC-labeled PTX + RTV − NMF solution, and the cells were incubated for 3, 6, 12, and 24 h. A time-dependent uptake of the two formulations was determined. The results are depicted in Figure 8. Figure 8A–C depict the uptake of HA − PTX + RTV − NMF and PTX + RTV − NMF in MCF-7, MDA-MB-231, and MCF-12A cells, respectively. The mean fluorescence intensity of HA − PTX + RTV − NMF and PTX + RTV − NMF was statistically higher than the corresponding control group from the 6 h time points onward in both the metastatic breast cancer cell lines and normal breast cell line. In all the three cell lines, at 6 h, there was a significant difference between HA-targeted nanomicelles and HA-nontargeted nanomicelles uptake. After 6 h, the uptake for both treatment groups was similar. Uptake and internalization of the nanomicelles plateaued 12 h onward. 

#### 2.9.2. Intracellular Distribution Using Confocal Microscopy

The intracellular distribution of HA − PTX + RTV − NMF was determined qualitatively by incubating FITC-labeled HA − PTX + RTV − NMF and FITC-labeled PTX + RTV − NMF in MCF-7, MDA-MB-231, and MCF-12A cells (Figure 9) for 3 and 6 h. The intracellular distribution of targeted nanomicelles was observed using confocal laser scanning microscopy (CLSM). As evident from Figure 9A,B, the intracellular uptake and distribution at 3 h was less than that at 6 h in MCF-7 cells. A similar trend is also evident for the MDA-MB-231, as seen in Figure 9C,D. The intracellular uptake of the two treatment groups was relatively less in MCF-12A cells (Figure 9E,F) as compared to MCF-7 and MDA-MB-231 cells. This can be due to the relatively lower expression of CD44 receptors on MCF-12A cells and lower levels of GSH, which induces the cleavage of the disulfide bond in HA − PTX + RTV − NMF. The nanomicellar formulation internalizes effectively with the cell by 6 h for all the three cell lines, and there is a significant difference between the HA-targeted and nontargeted nanomicelles. Nanomicelles primarily integrate with the cell membrane and get ingested in the cells via receptor-mediated endocytosis. This enables effective delivery of the hydrophobic cargo to the cell [5,18,19]. Interestingly, the uptake for MCF-7 and MDA-MB-231 cells treated with the targeted nanomicelles (HA − PTX + RTV − NMF) was higher than the nontargeted nanomicelles (PTX + RTV − NMF). MCF-7 and MDA-MB-231 cells express a plethora of CD44 receptors on their surface. The ligand for this receptor, HA, recognizes it and can result in a higher accumulation of the nanomicelles within the cells [23,24].

### 2.10. In Vitro Cytotoxicity Determination

The in vitro cytotoxicity of HA − PTX + RTV − NMF, PTX + RTV − NMF, and placebo NMF was evaluated in breast cancer cells MCF-7, MDA-MB-231, and normal breast epithelium MCF-12A cells. This study was performed to assess the cytotoxicity of the PTX nanomicellar formulation. Cell viability of the placebo NMF was also evaluated to analyze the effect of the polymers on cell viability. All the cells were treated with the various treatment groups such as (i) HA − PTX + RTV − NMF, (ii) PTX + RTV − NMF, and (iii) placebo NMF for 6 h, followed by removal of the treatment groups and addition of fresh complete media. This was followed by incubating the cells for 24 and 72 h and analyzing the cell viability using MTT assay (Figure 10). A 6 h time point for the treatment was chosen as at around 6 h, there is a significant difference between uptake and internalization of the targeted and nontargeted nanomicelles in all cell lines. In addition, the uptake at 12 and 24 h tends to saturate, and there is not much significant difference between the two nanomicelles treatment groups. Figure 10A depicts the cell viability (%) at 24 h for HA − PTX + RTV − NMF, PTX + RTV − NMF, and placebo NMF and Figure 10B for 72 h. All the three cell lines approximately demonstrate 80–85% cell viability for the placebo NMF at 24 and 72 h, indicating the safety of the polymers used for the formulation. Vit E-TPGS and PLGA polymers are generally regarded as safe (GRAS) polymers. HA is an endogenous component. It is an essential component of the extracellular matrix and an endogenous ligand for CD44 receptors. Cell viability for MCF-7 and MDA-MB-231 breast cancer cells was lower for HA-targeted nanomicelles as compared to the nontargeted nanomicelles. This can be due to increased uptake of the HA − PTX + RTV − NMF in the breast cancer cells and due to the reduction-triggered release of PTX due to the GSH enzyme. When comparing the cell viability between different cell lines, MCF-7 cells have a lower cell viability as compared to MDA-MB-231 cells. It can be concluded that MCF-7 cells are more sensitive to HA − PTX + RTV − NMF and PTX + RTV − NMF than MDA-MB-231 cells. This result clearly correlates the cellular uptake by FACS. In this study, FITC intensity was significantly lower in MDA-MB-231 cells compared to MCF7 cells after FITC-labeled PTX + RTV − NMF incubation, indicating a lower expression of CD44 in MDA-MB-231 cells. The normal cells (MCF-12A) demonstrate higher cell viability than the breast cancer cells MCF-7 and MDA-MB-231, specifically for the HA − PTX + RTV − NMF treatment group. This can be due to the lower density of CD44 receptors present on these cells and due to the lower GSH enzyme concentration in the cytoplasm. These factors in combination reduce the uptake of HA − PTX + RTV − NMF in the normal breast cells and decrease the release of PTX in the cytoplasm. The cell viability of PTX + RTV − NMF in MCF-12A cells is comparable to breast cancer cells. This can indicate that targeting with HA and stimuli-sensitive release by disulfide bonds in the nanomicelles can play a protective role in reducing off-target toxicities. This can also ensure higher drug release and selective targeting of the breast cancer cells by HA − PTX + RTV − NMF.

### 2.11. In Vitro Determination of Targeting Efficiency

To determine whether HA targeting can increase the intracellular uptake of HA − PTX + RTV − NMF, the breast cancer cells were treated with HA − PTX + RTV − NMF in the presence and absence of excess HA. MCF-7 and MDA-MB-231 cells were treated with excess HA. After 30 min of this treatment, HA − PTX + RTV − NMF was added to the cells. Control cells only received HA − PTX + RTV − NMF without externally added HA. The uptake was determined by CLSM, as seen in Figure 11. The fluorescence intensity was reduced in the groups where excess free HA was added, followed by the addition of HA − PTX + RTV − NMF. This was seen in both breast cancer cell lines MDA-MB-231 (Figure 11A) and MCF-7 (Figure 11B). This can indicate that HA − PTX + RTV − NMF is internalized in MCF-7 and MDA-MB-231 cells by receptor-mediated endocytosis.

### 2.12. Cellular Uptake by UHPLC-MS

Quantitative determination of PTX uptake in MCF-7, MDA-MB-231, and MCF-12A cells was performed using the UHPLC-MS method described in the Methods section for determining if the nanomicelles could effectively release PTX in the cells. This study was also performed to determine whether the RTV addition to the nanomicelles could effectively function as a P-gp inhibitor and increase the intracellular PTX concentration. The overexpression of MDR proteins such as P-gp efflux pump on the cell membrane of breast cancer cells plays an important role in the active efflux of hydrophobic chemotherapeutic agents [25]. This leads to an ineffective concentration on the drug inside the cells. To compensate this, the dose of the drugs is generally increased. This in turn leads to increased off-target effects and toxicity [26]. The uptake of PTX was compared in formulations like HA − PTX + RTV − NMF and HA − PTX − NMF, in both the cell lines separately. The uptake was observed at 6 and 12 h. Results are depicted in Figure 12. The uptake of PTX from HA − PTX + RTV − NMF was higher than the PTX uptake in the HA − PTX − NMF treatment group in MCF-7 cells (Figure 12A) at the 6 and 12 h time point. Similar results were also observed in MDA-MB-231 cells treated with HA − PTX + RTV − NMF and HA − PTX − NMF (Figure 12B). For MCF-12 A breast epithelium cells, at six hours, there was no significant difference between the two treatment groups, but at 12 h, the uptake of PTX from the HA − PTX + RTV − NMF formulation was significantly lower than the PTX + RTV − NMF group (Figure 12C). In addition, the total overall uptake of PTX in normal cells was lower than the breast cancer cells. This can signify the importance and effectiveness of the CD44-mediated selective uptake of HA − PTX + RTV − NMF and importance of RTV in inhibiting P-gp. This study can confirm the hypothesis that the addition of RTV to HA − PTX − NMF can result in a higher intracellular accumulation of PTX in the breast cancer cells like MCF-7 and MDA-MB-231.

### 2.13. In Vitro Potency Determination

#### 2.13.1. Mitochondrial Membrane Potential

Mitochondrial membrane potential (ΔΨm) is an effective biomarker to assess the functioning of the mitochondria [27]. Continuously dividing cancer cells demonstrate the elevated presence of mitochondria [28,29]. Loss of ΔΨm is regarded as an important hallmark of cell apoptosis [30,31,32]. The primary purpose of this study was to determine whether HA − PTX + RTV − NMF treatment could lower the ΔΨm in the breast cancer cells. This was compared to the nontargeted formulation PTX + RTV − NMF. Secondly, this study was also performed to see if the addition of RTV to the nanomicelles could increase the intracellular PTX concentration and thus cause reduction in ΔΨm. Therefore, this HA − PTX + RTV − NMF was compared to HA − PTX − NMF. Figure 12 depicts the ΔΨm of (Figure 13A) HA − PTX + RTV − NMF and PTX + RTV − NMF in MCF-7 and MDA-MB-231 cells, and (Figure 13B) HA − PTX + RTV − NMF and HA − PTX − NMF in MCF-7 and MDA-MB-231 cells. In Figure 13A, HA − PTX + RTV − NMF was able to release PTX intracellularly in both the breast cancer cells and cause reduction in ΔΨm. The reduction was significantly lower in the HA-targeted group than the nontargeted group. This can indicate that targeting with HA increases the intracellular uptake and subsequently releases more PTX intracellularly. In Figure 13B, the reduction in ΔΨm was significantly lower in the HA − PTX + RTV − NMF group as compared to the HA − PTX − NMF group for both the cell lines. This can indicate that RTV from the nanomicelles blocks the P-gp and slows the efflux of PTX from the cancer cells. This can lead to the possible reduction in MDR [33,34].

#### 2.13.2. Evaluation of Reactive Oxygen Species

The ability of HA − PTX + RTV − NMF to generate Reactive Oxygen Species (ROS) in breast cancer cell lines was evaluated by 2′,7′–dichlorofluorescin diacetate (DCFDA) assay. For this purpose, MCF-7 and MDA-MB-231 cells were treated with (i) HA − PTX + RTV − NMF and (ii) PTX + RTV − NMF to determine if its higher uptake and HA targeting can induce ROS in breast cancer cells. The same cells were also treated with (i) HA − PTX + RTV − NMF and (ii) HA − PTX − NMF to see if the addition of RTV to the formulation can block the efflux of PTX from the cancer cells by inhibiting P-gp. Figure 13C depicts the mean fluorescence intensity of MCF-7 and MDA-MB-231 cells treated with HA − PTX + RTV − NMF and PTX + RTV − NMF. Here, ROS generation in the breast cancer cells is significantly lower in the cell lines after 12 h of treatment. This can indicate that, because of the higher uptake of HA-targeted PTX+RTV nanomicelles, a higher amount of PTX was released in the cells. This led to the lowering of the production of ROS in the cancer cells. Additionally, both the targeted and nontargeted nanomicellar formulations were able to reduce the unhindered proliferation in the breast cancer cells, on similar lines, in Figure 13B.

## 3. Conclusions

In this study, a hyaluronic acid-targeted mixed-nanomicelles-encapsulating chemotherapeutic drug PTX and a P-gp inhibitor RTV was prepared and optimized (HA − PTX + RTV − NMF). Although numerous nano-formulations for cancer drug delivery have been developed, very few have led to successful clinical transition. This is due to suboptimal therapeutic outcomes possibly caused by inefficient drug release, effective drug intracellular accumulation, and off-target toxicity of nanocarriers [35]. This nanomicellar formulation had a size below 200 nm which can lead to effective EPR effect in the tumor tissue. Further, a size lower than 200 nm can be useful to evade recognition by the reticuloendothelial system (RES), sparing phagocytosis of the nanomicelles by macrophages. The nanomicelles were constructed from polymers like HA−ss−PLGA and Vitamin E TPGS. HA − PTX + RTV − NMF demonstrated serum stability, temperature stability, and effective drug release in a reductive environment. Drug resistance to the chemotherapeutic agent, mainly acquired due to over-burdening and excessive doses of chemotherapy, can lead to increasing efflux pumps like P-gp expression on the cancer cells [36,37,38,39]. Once the chemotherapeutic agents (e.g., paclitaxel, doxorubicin, vincristine) are released from the nanocarriers, these are refluxed by the efflux transporters like P-gp, MRP, and BCRP. This leads to a decrease in drug accumulation in the cancer cells to sub-therapeutic concentrations. Consequently, to make the chemotherapy effective, clinicians either increase doses or use combinations of many drugs, causing too much toxicity to healthy cells.

Here, we developed a combination nanomicellar approach for delivering an anti-cancer drug and a drug that will lower the chances of MDR in breast cancer cells. Here, we utilized bio-inspired and “smart” nanomicelles having HA targeting. The bio-inspired nature of the nanomicelles comes from the fact that HA is a component of the extracellular matrix and a ligand for CD44 receptors. HA is bound to PLGA polymer with disulfide bonds, which are reduced in the GSH-rich cancer intracellular environment. This stimulus (GSH-induced nanomicellar disintegration) drastically slows down the release of the drug in healthy cells, as seen in the cytotoxicity study. With an effective targeting moiety and stimuli-responsive nature of the nanomicelles, the release of cytotoxic PTX in MCF-7 and MDA-MB-231 breast cancer cell lines was achieved in this research study. The addition of RTV; a potent MDR modulator and P-gp inhibitor-prevents efflux and allows increased intracellular uptake and retention of PTX, as seen by the uptake determination study by UHPLC-MS. The HA − PTX + RTV − NMF formulation demonstrated a higher accumulation of PTX in breast cancer cells (MCF-7 and MDA-MB-231) compared to MCF-12A (noncancer breast cells). HA is a natural ligand for CD44 receptors, which are overexpressed in breast cancer cells like MCF-7 and the triple-negative breast cancer cell line MDA-MB-[40,41,42,43]. HA − PTX + RTV − NMF enters through the receptor-mediated endocytosis process in the breast cancer cells [44]. Cancer cells overexpress CD44 and allow HA − PTX + RTV − NMF absorption, and consequently, more accumulation selectively in MCF-7 and MDA-MB-231 compared to MCF-12A. HA − PTX + RTV − NMF contains a disulfide bond (HA-ss-PLGA) and the drug is released in the presence of GSH, which is abundant in cancer cells. It was also effective in reducing the mitochondrial membrane potential and decreasing the reactive oxygen species in breast cancer cells. HA − PTX + RTV − NMF can deliver the anticancer agent to targeted cancer cells avoiding normal cells, as reflected in the cell cytotoxicity assay where MCF-7 cells survived significantly less compared to MCF12-A cells. Thus, it can be concluded that the combination of the targeted reduction-sensitive release of PTX by the HA−ss−PLGA polymer and reversal of MDR by RTV appears to be an effective strategy for anticancer drug delivery.

## 4. Materials and Methods

### 4.1. Materials

Active pharmaceutical ingredients used here like PTX and RTV were purchased from LC laboratories, Massachusetts USA. Polymers like vitamin E TPGS, dichloromethane (DCM), ethanol (EtOH), acetonitrile (AcN), and HPLC-grade water (H_2_O) were purchased from Thermo Fisher Scientific, Waltham, MA, USA. Distilled deionized (DDI) water was obtained from Barnstead EasyPure UV deionization systems (Barnstead|Thermolyne, Dubuque, IA, USA). The following materials Dulbecco’s modified Eagle’s medium (DMEM) and trypsin (TrypLE) were from Invitrogen (Carlsbad, CA, USA), nonessential amino acids were purchased from Molecular Probes by Life Technologies^®^ (Thermo Fisher Scientific, Waltham, MA, USA). Fetal bovine serum (FBS) from Atlanta Biologics (Lawrenceville, GA, USA). Fluorescein isothiocyanate (FITC) reagent from Molecular Probes by Life Technologies^®^ (Thermo Fisher Scientific, Waltham, MA, USA). Other chemicals and reagents used in this study were purchased from Sigma-Aldrich (St. Louis, MO, USA) unless specified.

### 4.2. Cell Culture

MCF-7, MCF-12A, and MDA-MB-231 cells were obtained from American Type Culture Collection (ATCC, Manassas, VA, USA). All cell lines were stored in a liquid nitrogen tank, which was maintained at −198 °C until further use. The cells were cultured in a T-75 Corning flask purchased from Fisher Scientific. Cells were cultured in DMEM (Gibco, Gaithersburg, MD, USA) containing 10% serum (heat inactivated FBS), antibiotics (100 IU/mL of Streptomycin and 100 IU/mL of penicillin), 1% sodium pyruvate, and 1% of nonessential amino acids. The cells were maintained at 37 °C (normal body temperature), 5% CO_2_, and 90% relative humidity in a Heracell 150 CO_2_ incubator (Marshall Scientific, Hampton, NH, USA). The cells were harvested when they reached 80–90% confluence. The cell growth was observed under a ZEISS Telaval 31 Inverted Phase Contrast Microscope.

### 4.3. Synthesis of HA-ss-PLGA Graft Copolymer

HA-ss-PLGA graft copolymer was synthesized in three steps; first, PLGA polymer was functionalized with *N*-hydroxysuccinimide (NHS); second, hyaluronic acid (HA) was functionalized with cysteine (CYS); and lastly, functionalized PLGA−NHS and HA−CYS were reacted to form HA−ss−PLGA graft copolymer. The section below describes the detailed synthesis of each of the polymers and their characterization by H^1^ NMR spectroscopy.

For the synthesis of PLGA−NHS polymer, PLGA copolymer was synthesized by ring-opening polymerization [45,46]. Briefly, L-lactide and glycolide were weighed and transferred to a round-bottom flask. The polymerization reaction was catalyzed by the addition of stannous octate [Sn (Oct)_2_ (0.05%, *w*/*w*)]. The reaction was carried out at 150 °C for 12 h in an N_2_ environment. The resulting polymer was dissolved in anhydrous dichloromethane (CH_2_Cl_2_) and transferred to a 50 mL centrifuge tube. The polymer was precipitated by the addition of excess cold diethyl ether. The resulting precipitated poly (lactic-co-glycolic acid) (PLGA) polymer was centrifuged and washed three times with cold diethyl ether. The final polymer formed was evaporated at a high speed under vacuum (Genevac, Ipswich, Suffolk, UK). The polymer was analyzed by H^1^ NMR spectroscopy to confirm its formation. Further, this synthesized PLGA was functionalized with NHS using ethyl-3-(3-dimethylaminopropyl)-carbodiimide (EDC)/N−hydroxysuccinimide (NHS) chemistry. Briefly, 1.0 g PLGA was dissolved in anhydrous CH_2_Cl_2_. This was followed by adding 0.40 EDC and 0.24 g of NHS. The reaction was conducted at room temperature (22 °C) for 24 h with constant stirring. The PLGA−NHS polymer formed was precipitated using cold diethyl ether, washed three times with cold diethyl ether, and centrifuged for separation. PLGA−NHS was finally dried at high speed under vacuum (Genevac, Ipswich, Suffolk, UK).

Hyaluronic acid (HA) functionalized with cystamine dihydrochloride (CYS) was synthesized (HA−CYS) using EDC chemistry. CYS can act as a disulfide linker donating molecule for forming disulfide bonds (−ss−) in antibody-drug conjugates, polymer-drug conjugates, polymer-peptide conjugates, and polymer-polymer conjugates [47]. Briefly, 5.0 mmol HA, 4.5 mmol EDC, and 5.0 mmol CYS were dissolved in PBS (pH = 6.8). The reaction was continuously stirred and allowed to proceed for 24 h under an N_2_ environment. After the reaction, the product was dialyzed against DDI water using a Slide-A-Lyzer dialysis cassette (MW 2000 kDa). The formed product HA−CYS was lyophilized until further use. Lastly, to synthesize HA-ss-PLGA graft copolymer, the reaction products of the above two reactions were combined. Briefly, 1.0 g of PLGA-NHS was dissolved in 20 mL anhydrous DMF, and 200.0 mg of HA-CYS was added. The reaction was proceeded at room temperature (22 °C) for 48 h with constant stirring. This was followed by dialyzing the final product against DDI water using the Slide-A-Lyzer dialysis cassette (MW 2000 kDa) overnight and lyophilized to obtain the final HA−ss−PLGA graft copolymer.

### 4.4. Critical Micellar Concentration

The two polymers used in this study, Vit E-TPGS and HA−ss−PLGA, are amphiphilic block polymers. These polymers spontaneously form micelles in an aqueous medium above a certain concentration called the critical micellar concentration (CMC). The CMC of Vit E-TPGS, HA−ss−PLGA, and a mixture of Vit E-TPGS and HA−ss−PLGA (Vit E-TPGS: HA−ss−PLGA::4:2) was determined using hydrophobic iodine as a probe. Nanomicellar formulations of varying concentrations of Vit E-TPGS (6.51 × 10^−9^ − 7 wt%), HA−ss−PLGA (3.72 × 10^−9^ − 2 wt%), and a mixture of Vit E-TPGS and HA−ss−PLGA (1.8 × 10^−9^ − 8 wt%) were prepared using serial dilutions. A solution of iodine (I_2_) and potassium iodine (KI) in the ratio I_2_:KI::0.5:1, was mixed in DDI water. Then, 100 µL NMF was added to a 96-well plate. After, 15 µL of I_2_:KI solution was added to each NMF, followed by incubation for 5 h in the dark at room temperature. The absorbance of hydrophobic iodine partitioned into the hydrophobic nanomicellar core was determined by a microplate absorbance spectrophotometer (BioRad, Hercules, CA, USA). The absorbance was recorded at two emission wavelengths: 286 and 460 nm.

### 4.5. Design of Experiment for Nanomicelles Preparation

Nanomicelles for hydrophobic drugs like PTX and RTV were prepared by using polymers Vit E-TPGS and HA−ss−PLGA. The formulation was designed by employing a Design of Experiment (DOE) protocol to study the effect of changing variables on nanomicellar formulation quality outcomes. In the DOE, a set of independent and dependent variables were selected. Polymer concentrations of Vit E-TPGS (wt%) and HA−ss−PLGA (wt%), and sonication time (min) were selected as independent variables, while nanomicellar size, zeta potential, polydispersity index, drug-loading efficiency, and drug-entrapment efficiency for both the drugs, PTX and RTV, were set as dependent variables. The student version of JMP^®^ 10.0 software was employed for the DOE and data analysis. The concentrations of drugs, PTX and RTV, were kept fixed in the nanomicellar formulations. A full factorial design of the experiment was selected to analyze various formulations having a fixed combination of independent variables. Three independent and seven dependent variables were identified in the DOE. The independent variables were X1: sonication time (min), X2: HA-ss-PLGA (wt%), and X3: Vit E-TPGS (wt%), while the dependent variables were Y1: size (nm), Y2: polydispersity index (PDI), Y3: zeta potential (mV), Y4: PTX drug entrapment efficiency (EE) (%*w*/*w*), Y5: PTX drug loading efficiency (LE) (%*w*/*w*), Y6: RTV drug entrapment efficiency (EE) (%*w*/*w*), and Y7: RTV drug loading efficiency (LE) (%*w*/*w*). The full factorial design of experiments gave rise to 11 combinations or runs of independent variables for analyzing their effect on dependent variables. For each independent variable, higher (1), middle (0), and lower (−1) values were selected. The final output of all the DOE is summarized in Table 1.

### 4.6. Preparation of Nanomicellar Formulation

Hyaluronic acid-targeted, PTX-and RTV-loaded nanomicellar formulation (HA − PTX + RTV − NMF) was prepared by the solvent evaporation-film rehydration method as described previously [5,43,44]. Briefly, 1.0 mg of PTX and 0.25 mg of RTV were weighed accurately and dissolved in 5 mL of DCM ethanol mixture (DCM:ethanol::2:1). To this mixture of drugs, Vit E-TPGS and HA−ss−PLGA were weighed separately according to various formulations listed in Table 1 and dissolved in the above drug solution. The organic solvent was evaporated at a high speed under vacuum (Genevac, Ipswich, Suffolk, UK) for approximately 10 h to obtain a solid film. The resulting film was rehydrated with HPLC-grade water. Briefly, 1.0 mL of HPLC-grade water was added to the solid film. The formulation was sonicated using a bath sonicator until the film completely dissolved and a final HA − PTX + RTV − NMF was obtained. The formulation was filtered through a 0.22 µm nylon syringe filter (Tisch Scientific, North Bend, OH, USA). This ensured removal of any foreign particles and polymer agglomerates to obtain a uniform nanomicellar size distribution. In a similar method, untargeted PTX + RTV − NMF was also prepared from combining drugs PTX+RTV in the same amounts as mentioned above and using polymers like PLGA and Vit E TPGS. HA − PTX + RTV − NMF, PTX + RTV − NMF, and all nanomicellar drug formulations were stored at 4 °C until further use.

#### 4.6.1. Formulation Characterization: Size, Morphology, Zeta Potential

The critical quality attributes (CQAs) of HA − PTX + RTV − NMF such as hydrodynamic size, polydispersity index (PDI), and zeta potential were determined by a dynamic light scattering (DLS) instrument (Zetasizer Nano ZS, Malveran Instrument Ltd., Worcestershire, UK). For this, 1.0 mL of HA − PTX + RTV − NMF was placed in a glass cuvette. The instrument was calibrated to measure three values of hydrodynamic size and PDI for each formulation. The size was measured in nanometers (nm) and PDI was measured as a unitless number. An average of three values for size and PDI was calculated for each sample. For determination of zeta potential, DTS1060 glass cuvettes were used. Similar to the size and PDI, an average of three values was considered as the final zeta potential for each formulation. The zeta potential was measured in millivolts (mV). The morphological characteristics of HA − PTX + RTV − NMF were visualized using transmission electron microscopy (TEM). A single drop of HA − PTX + RTV − NMF was placed on the copper grid of the microscope. A layer of nitrocellulose and carbon was applied in the evaporator, which was stained by 1% uranyl solution. The photographs were taken by a JEM 1200 EX II TEM at a voltage of 100 kV.

#### 4.6.2. Drug-Entrapment and Drug-Loading Efficiency

Method development: The drug entrapment and loading efficiencies for PTX and RTV entrapped in the nanomicelles were determined simultaneously by the UHPLC-MS method. A simultaneous quantification of PTX and RTV was performed in a positive mode using a QTrap^®^ API-3200 (Sciex, Framingham, MA, USA) Mass spectrometer. Electrospray ionization was used to generate ions of analytes like PTX and RTV. A reverse-phase C-18 column (Kinetex^®^ 4.6 µm, C18 100Å, 100 × 2.30 mm, Phenomenex Inc., Torrance, CA, USA) ran at 0.2 mL/min with mobile phase A = water (H_2_O) with 0.1% formic acid (FA) and mobile phase B = acetonitrile (ACN) with 0.1% FA in a gradient. The UV/Vis detector was set at 220 nm wavelength. The retention time for PTX and RTV was 13.7 and 14.2 min, respectively. In addition, the Q1/Q3 masses for PTX and RTV were 854.42/105.10 and 721.45/140.30, respectively. A standard curve of the PTX and RTV drug was obtained by injecting varying concentrations of the mixture of the drugs in the ratio PTX:RTV::4:1, as this was the ratio of the two drugs used in the nanomicellar formulation. The concentration of PTX was 500.0–1.95 µg/mL and that of RTV was 125.00–0.49 µg/mL at an injection volume of 10 µL.

Sample preparation: 1.0 mL of HA − PTX + RTV − NMF was centrifuged at 100,000 rpm for 10 min. The supernatant was collected and transferred into new Eppendorf tubes. The formulation was lyophilized to obtain a solid white residue containing polymers and drugs PTX and RTV. One milliliter of DCM was added to the lyophilized product and vortexed to obtain a clear solution. The addition of nonaqueous solvent like DCM reverses HA − PTX + RTV − NMF, dissolving the hydrophobic drug. DCM was evaporated under vacuum using vacuum (Genevac, Ipswich, Suffolk, UK). The solid pellet of polymers and drugs obtained was resuspended using 1.0 mL of mobile phase. The entrapment and loading efficiencies of formulation F1–F11 prepared in triplicates were determined by the following equations:(1)$$\frac{\mathrm{Entrapment}\;\mathrm{Efficiency}\;=\;\left(\mathrm{amount}\;\mathrm{of}\;\mathrm{PTX}/\mathrm{RTV}\right)\;\times \;100\;}{\left(\mathrm{amount}\;\mathrm{of}\;\mathrm{PTX}/\mathrm{RTV}\;\mathrm{added}\;\mathrm{in}\;\mathrm{HA}-\mathrm{PTX}+\mathrm{RTV}-\mathrm{NMF}\;\right)}$$
(2)$$\frac{\mathrm{Entrapment}\;\mathrm{Efficiency}\;=\;\left(\mathrm{amount}\;\mathrm{of}\;\mathrm{PTX}/\mathrm{RTV}\right)\;\times \;100\;}{\left(\mathrm{amount}\;\mathrm{of}\;\mathrm{PTX}/\mathrm{RTV}\;\mathrm{added}\;\mathrm{in}\;\mathrm{HA}-\mathrm{PTX}+\mathrm{RTV}-\mathrm{NMF}\;+\;\mathrm{polymers}\;\mathrm{added}\right)}$$

### 4.7. Nanomicellar Dilution Study

The effects of dilution on size, zeta potential, and PDI were determined using DLS. Briefly, HA − PTX + RTV − NMF was diluted up to 200 times using HPLC-grade water. The diluted nanomicelles were stored for 24 h at two temperature conditions, one at room temperature (RT) and the other at 37 °C. One milliliter of the resultant diluted HA − PTX + RTV − NMF was analyzed for change in nanomicellar size.

### 4.8. Dissolution and Drug Release

Dissolution and drug release of PTX and RTV from HA − PTX + RTV − NMF were determined by the UHPLC-MS method as mentioned above. Briefly, 1.0 mL of HA − PTX + RTV − NMF was transferred to a dialysis tubing of a cut-off molecular weight of 2000 kDa (Spectrum Laboratories, Irving, TX, USA). HA − PTX + RTV − NMF and PTX + RTV − NMF were suspended in various buffers as (i) PBS at pH 7.4, (ii) PBS at pH 6.8, (iii) PBS with 10% FBS at pH 7.4, and (iv) PBS with 10% FBS at pH 6.8. The dialysis tubing was surrounded externally by 5 mL of the buffer solution in a 15 mL centrifuge tube. Then, 1.0 mL samples of the external fluid were collected at predetermined time points, evaporated using vacuum, and then resuspended in the mobile phase. The external buffer solution was replaced with 1.0 mL of fresh buffer to maintain sink conditions. The concentration of PTX released in the buffer solution was calculated.

### 4.9. Stability Studies

Storage stability and plasma stability studies for HA − PTX + RTV − NMF and PTX + RTV − NMF were conducted by analyzing the size, PDI, and zeta potential of the nanomicelles. Storage stability involved storage of HA − PTX + RTV − NMF and PTX + RTV − NMF at temperatures 4 °C, 22 °C and 37 °C. Regular samples were withdrawn at Day 0, Day 3, and Day 7. For plasma stability studies, the nanomicellar formulations were stored in 10% FBS in DPS solution and stored at body temperature, 37 °C, in a water bath with constant shaking to simulate physiological conditions. Here, 1.0 mL of sample was withdrawn at Day 2, Day 5, and Day 7. The stability studies were assessed by measuring the HA − PTX + RTV − NMF and PTX + RTV − NMF. CQA’s size, PDI, and zeta potential were determined at these conditions.

### 4.10. Nanomicelles in Reduction Stimulated Environment

Targeted (HA − PTX + RTV − NMF) and nontargeted (PTX + RTV − NMF) nanomicellar formulations were mixed with equal volumes of 50 mM GSH in PBS. A specific concentration of 50 mM GSH was selected as it represents the amount of GSH present in cancer cells cytoplasm [48,49]. The formulations were stored in a water bath maintained at 37 °C with shaking at 100 rpm. Particle size of the formulation was analyzed at predetermined time points using DLS. As a control, the nanomicellar formulations were stored in a similar PBS solution without GSH.

A cumulative amount of PTX release from the above samples was determined. For this, HA − PTX + RTV − NMF and PTX + RTV − NMF were incubated with 50 mM GSH in PBS for 48 h in a water bath at 37 °C. Incubation in PBS at the same conditions was considered as control. The samples were centrifuged at 100,000 rpm for 20 min to separate the nanomicelles. The supernatant containing the dissolved drug was collected. The supernatant was filtered and dried using vacuum. The film was rehydrated with the mobile phase and the samples were analyzed by the UHPLC-MS method stated above.

### 4.11. Cellular Uptake and Intracellular Distribution Study

The cellular uptake and intracellular distribution of HA − PTX + RTV − NMF were determined in breast cancer cell lines like MCF-7 and MDA-MB-231 and normal breast epithelium cells like MCF-12A. HA − PTX + RTV − NMF was labeled with FITC. The nontargeted nanomicelles, PTX + RTV − NMF, were also labeled with FITC to see the effectiveness of the targeting agent, HA, in HA − PTX + RTV − NMF as compared to the nontargeted PTX + RTV − NMF. The cellular uptake was determined by analyzing the various treated cell lines by fluorescence-assisted cell sorting (FACS), and the intracellular distribution was determined with confocal laser scanning microscopy (CLSM).

#### 4.11.1. FITC Labeling

In vitro uptake of HA − PTX + RTV − NMF and PTX + RTV − NMF was determined by FITC labeling. FITC is widely used as a florescent marker to label proteins, drugs, as well as polymers in this case. FITC is sparingly soluble in water; hence, FITC was dissolved in DMSO (1.0 mg/mL). HA − PTX + RTV − NMF and PTX + RTV − NMF solution were added to the above-made FITC solution. These solutions were incubated at 4 °C in the dark for 12 h. After incubation, 1 mL of 50 mM NH_4_Cl was added to inactivate unreacted FITC. The final solution of FITC-labeled HA − PTX + RTV − NMF and PTX + RTV − NMF was aliquoted and stored at −20 °C until further use.

#### 4.11.2. Cellular Uptake Study by FACS

The uptake of FITC-labeled HA − PTX + RTV − NMF and PTX + RTV − NMF in breast cancer cell lines like MCF-7 and MDA-MB-231 was determined by measuring samples by FACS. Briefly, MCF-7 and MBA-MD-231 cells were seeded in a 24-well plate with 5 × 10^4^ cells/well. Then, 2.0 mL of complete DMEM with 10% FBS was added in each well for the cell growth. The cells were incubated overnight. The medium was removed and replaced with 1.0 mL of serum-free medium (SFM). Next, 20 µL of FITC-labeled HA − PTX + RTV − NMF and FITC-labeled PTX + RTV − NMF was added to the respective wells. Both MCF-7 and MBA-MD-231 received the following three treatments: Control (only SFM), FITC − PTX + RTV solution, and FITC − HA − PTX + RTV solution. The cells were incubated for specific time points of 3, 6, 9, 12, and 24 h to determine the uptake of various formulations. At each time point, the media were removed, and the cells were washed with DPBS. Cells were detached with the help of 200 µL mL of trypsin. Cells were collected in a FACS tube and centrifuged at 20,000 rpm for 5 min. The supernatant was discarded, and the pellet was washed twice with DPBS. The final sample preparation was made in DPBS for FACS. The mean fluorescence intensities of FITC − PTX + RTV NMF and FITC − HA − PTX + RTV NMF and the control group were determined by FACS at an excitation wavelength of 490 nm.

#### 4.11.3. Intracellular Distribution Using Confocal Microscopy

The cellular uptake of FITC − PTX + RTV − NMF and FITC − HA − PTX + RTV − NMF was determined in MCF-7, MBA-MD-231, and MCF-12A cells in a time-dependent manner. FITC labeling was carried out as described above. Both MCF-7 and MBA-MD-231 cells were seeded at 1 × 10^4^ cells/well and were seeded in an 8-chamber confocal microscopy slide (Nunc Lab-Tek 8 chambered, Thermo Fisher Scientific, Waltham, MA, USA). The cells were supplemented with 200 µL complete DMEM containing 10% FBS. In addition, 10.0 µL of FITC − PTX + RTV NMF and FITC − HA − PTX + RTV NMF was added to the respective chambers. At each time point, the culture media were removed followed by washing of the cells three times with DPBS (3 × 5 min). Washing ensured removal of unabsorbed NMF. The washing step was followed by the fixation step. Cold 4% buffered paraformaldehyde solution (200 µL) was added to each well. The cells were incubated at 37 °C for 20 min for fixation. After 20 min, buffered paraformaldehyde solution was removed, and the cells were again washed with 200 µL DPBS (3 × 5 min). This was followed by staining the cells with mounting media containing DAPI for 15 min (Vectashield Antifade Mounting Medium). The cells were covered with a cover slip. All the cover slip-ends were sealed to prevent cellular dehydration and evaporation of mounting media. The confocal microscopy slides were stored at 4 °C until analyzed further. The cells were observed under Leica Confocal Laser Scanning Microscopy (Leica TCS SP5, Wetzlar, Germany).

### 4.12. Cytotoxicity Study

Cellular cytotoxicity of HA − PTX + RTV − NMF, PTX + RTV − NMF, and placebo NMF was determined in breast cancer cells MCF-7 and MBA-MD-231 and in normal breast epithelium cells like MCF-12A. The in vitro cytotoxicity was determined by 3-(4,5-dimethylthiazol-2-yl)-2,5-diphenyltetrazolium bromide (MTT) cell viability assay. MTT is a yellow tetrazole dye. Healthy living cells reduce yellow MTT dye to purple formazan crystals. All the cell lines were seeded in 96-well plates at a cell density of 1 × 10^4^ cells/well. The cells were then incubated overnight at 37 °C in a 5% CO_2_ environment and 90% relative humidity. The in vitro cytotoxicity for HA − PTX + RTV − NMF, PTX + RTV − NMF, and placebo NMF formulations was determined. Then, 10 µL of the sample containing 1.0 mg/mL PTX was added to the cells. The optimized F-1 formulation was used as the placebo NMF group. The treatment was terminated after 6 h and the cells were provided with fresh complete media. Cell viability was determined after 24 and 72 h after the initial treatment. After incubation, the medium was aspired, and the cells were washed with PBS twice. MTT stock solution was prepared by adding MTT reagent A and MTT reagent B in the ratio 100:1. In addition, 20 µL of MTT reagent was added to each well. The cells were incubated for 3 h in an incubator. Absorbance of formazan products was measured using a microplate reader (BioRad, Hercules, CA, USA) at an excitation wavelength of 485 nm. Triton-X 5% prepared in serum-free medium (SFM) served as the positive control, and blank SFM without any treatment groups served as the negative control. The cell viability was calculated according to the formula:$$\frac{\mathrm{Cell}\;\mathrm{Viability}=\left(\mathrm{Absorbance}\;\mathrm{of}\;\mathrm{sample}-\mathrm{absorbance}\;\mathrm{of}\;\mathrm{negative}\;\mathrm{control}\right)\times 100}{\left(\mathrm{Absorbance}\;\mathrm{of}\;\mathrm{positive}\;\mathrm{control}-\mathrm{absorbance}\;\mathrm{of}\;\mathrm{negative}\;\mathrm{control}\right)}$$

### 4.13. Cellular Uptake and P-gp Inhibition Determination by UHPLC-MS

For determination of PTX uptake, UHPLC-MS quantification of PTX in breast cancer cells was employed. This study also determined if the addition of RTV, a P-gp inhibitor, to HA − PTX + RTV − NMF could elevate the intracellular concentration of PTX. Hence, two nanomicellar formulation treatment groups were used in this study: (i) HA − PTX + RTV − NMF and (ii) HA − PTX − NMF, one containing RTV and the latter without RTV. Briefly, MCF-7 and MDA-MB-231 cells were seeded in 12-well plates at a density of 5 × 10^4^ cells/well. After overnight incubation, the cells were treated with 20 µL of HA − PTX + RTV − NMF or HA − PTX − NMF solution for 6 and 12 h. After each time-point, the cells were washed twice with PBS and once with cold PBS to terminate the uptake of the treatment groups in the cells. This was followed by the addition of acetonitrile to the cells for lysis by disruption of the cell membrane. This ensured that the trapped PTX drug could not be released from the cytoplasm into the external solution. Cell debris from the suspension was separated by centrifugation at 10,000 rpm for 20 min. The supernatant containing the dissolved drugs was separated and evaporated under vacuum (Genevac, Ipswich, Suffolk, UK). The resultant drug residue was reconstituted using the mobile phase, filtered, and analyzed for PTX using the UHPLC-MS method.

### 4.14. In Vitro Potency Determination

#### 4.14.1. Mitochondrial Membrane Potential Determination

The mitochondrial membrane potential (ΔΨm) of metastatic breast cancer cells treated with HA − PTX + RTV − NMF was determined using the JC-1 Mitochondrial Membrane Potential Flow Cytometry Assay Kit (Cayman Chemicals, Ann Arbor, MI, USA) [50]. Briefly, MCF-7 and MDA-MB-231 cells were seeded at a density of 5 × 10^5^ cells/well in a 12-well plate. This was followed by treating the cells with (i) HA − PTX + RTV − NMF, (ii) PTX + RTV − NMF, and the (iii) PTX drug for 24 h. The cells were stained with the JC-1 staining solution and acquired by FACS according to the manufactures protocol. The change in the mitochondrial membrane potential of the cells was calculated by the following equations:$$\frac{\Delta \psi \;m\;=\;\mathrm{Red}\;\left(J\;-\;\mathrm{aggregates}\right)\;\times \;100}{\mathrm{Green}\;\left(\mathrm{JC}\;-\;1\mathrm{monomer}\right)}$$
$$\frac{\mathrm{Relative}\;\Delta \psi \;=\;\mathrm{Sample}\;\Delta \;\psi \;m\;\times \;100}{\mathrm{Control}\;\Delta \;\psi \;m}$$

#### 4.14.2. Evaluation of Reactive Oxygen Species

The ability of HA − PTX + RTV − NMF to generate ROS in breast cancer cell lines was evaluated by DCFDA assay (Abcam, DCFDA cellular ROS Detection Assay Kit). DCFDA is a fluorogenic dye. For this purpose, MCF-7 and MDA-MB-231 cells were treated with (i) HA − PTX + RTV − NMF and (ii) PTX + RTV − NMF to determine if higher uptake and HA targeting can induce ROS in breast cancer cells. The same cells were also treated with (i) HA − PTX + RTV − NMF and (ii) HA − PTX − NMF to see if addition of RTV to the formulation could block the efflux of PTX from the cancer cells by inhibiting P-gp. The cells were seeded at a density of 0.5 × 10^5^ in a 12-well plate and treated for 24 h. Cells were harvested by trypsin addition. This was followed by addition of complete DMEM media containing FBS, and the cells were collected in FACS tubes. The cells were centrifuged at 20,000 rpm for 5 min and washed with 1.0 mL DPBS twice. DCFDA dye was finally added to the cells and the cells were incubated for 30 min in the dark. The cells suspension was acquired by flow cytometry.

### 4.15. Statistical Analysis

Significance in each study was determined by using at least three replicates for each experiment. The data were represented as mean ± standard deviation (SD). Statistical significance between different study groups was analyzed by two-way ANOVA or Student’s *t*-test. *p* < 0.05 indicated statistical significance in all experiments.

## Figures and Tables

**Figure 1 ijms-22-01257-f001:**
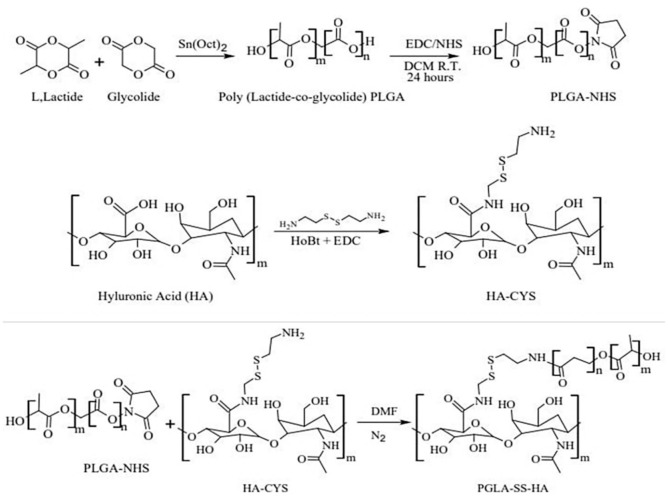
Synthesis scheme for hyaluronic acid-targeted poly (lactide) co (glycolide) (HA-ss-PLGA) graft co-polymer.

**Figure 2 ijms-22-01257-f002:**
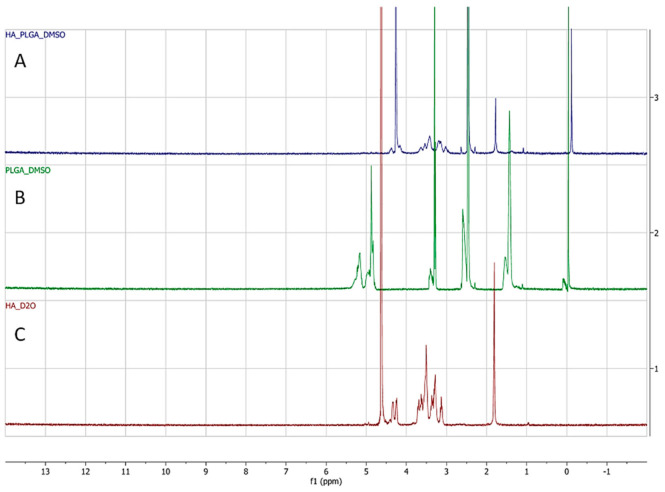
H^1^ NMR spectrum for (**A**) HA-ss-PLGA graft co-polymer dissolved in DMSO, (**B**) PLGA polymer dissolved in DMSO, and (**C**) HA dissolved in D_2_O.

**Figure 3 ijms-22-01257-f003:**
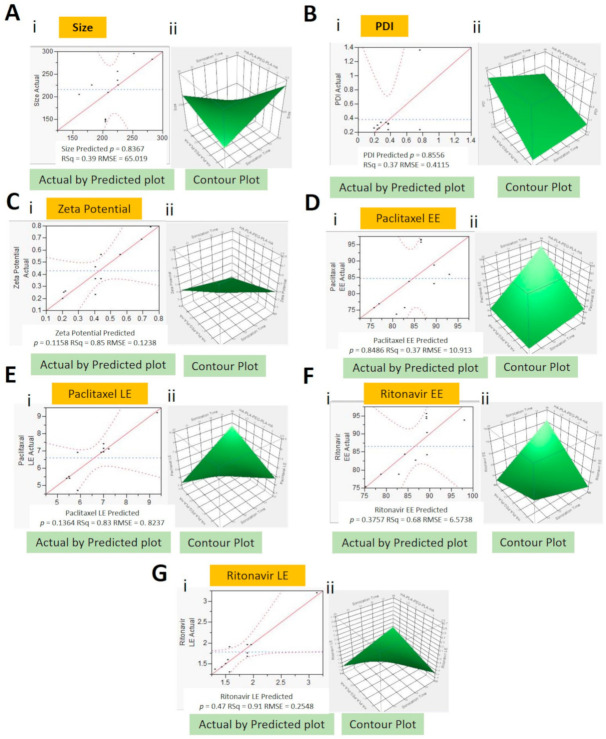
Actual by Predicted plots (i) and contour plots (ii) for dependent variables: Size (**A**), PDI (**B**), zeta potential (**C**), PTX entrapment efficiency (EE) (**D**), PTX loading efficiency (LE) (**E**), RTV EE (**F**), and RTV LE (**G**).

**Figure 4 ijms-22-01257-f004:**
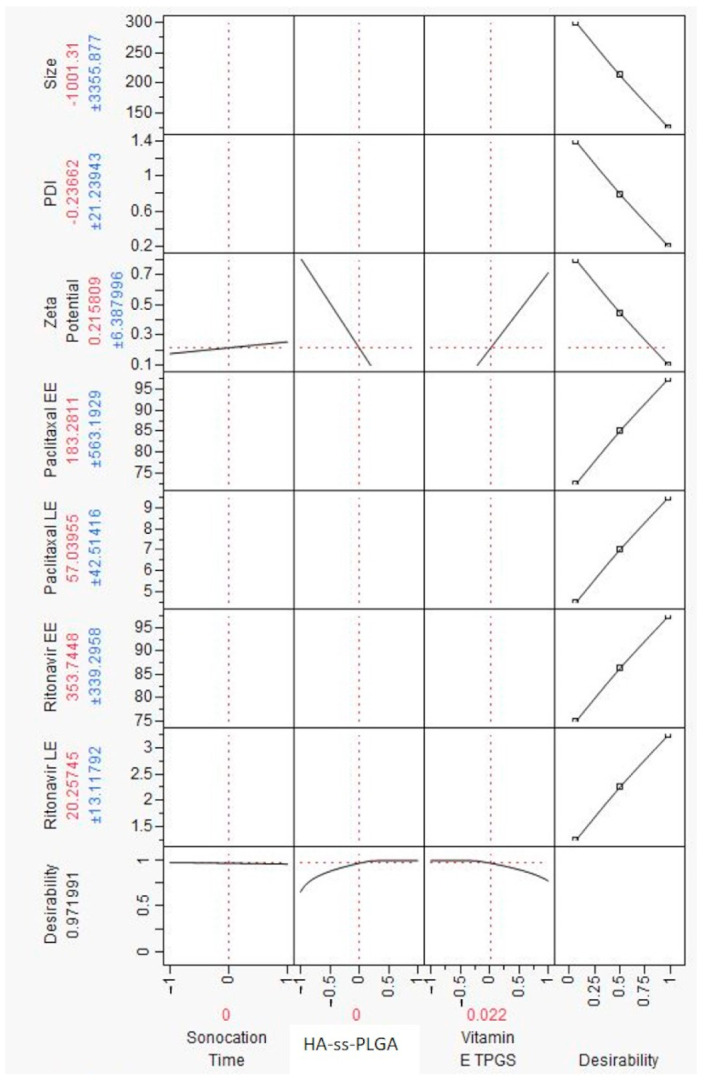
Prediction profiler for dependent variables of size, PDI, zeta potential, loading efficiency (LE), and entrapment efficiency (EE) depending upon the values of independent variables of polymer concentrations, HA−ss−PLGA, Vitamin E TPGS, and sonication time.

**Figure 5 ijms-22-01257-f005:**
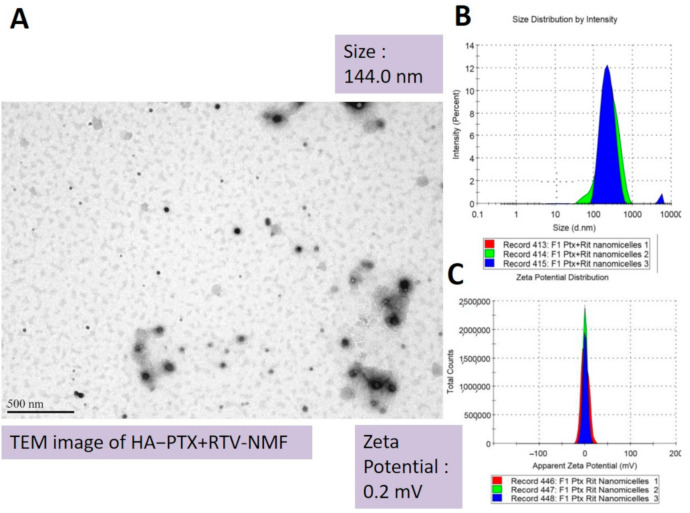
Morphological characteristics for hyaluronic acid targeted paclitaxel and ritonavir nanomicelles (HA − PTX + RTV − NMF). (**A**). Transmission electron microscopy (TEM) image, (**B**) size distribution by intensity (nm), (**C**) and apparent zeta potential (mV) of optimized HA − PTX + RTV − NMF.

**Figure 6 ijms-22-01257-f006:**
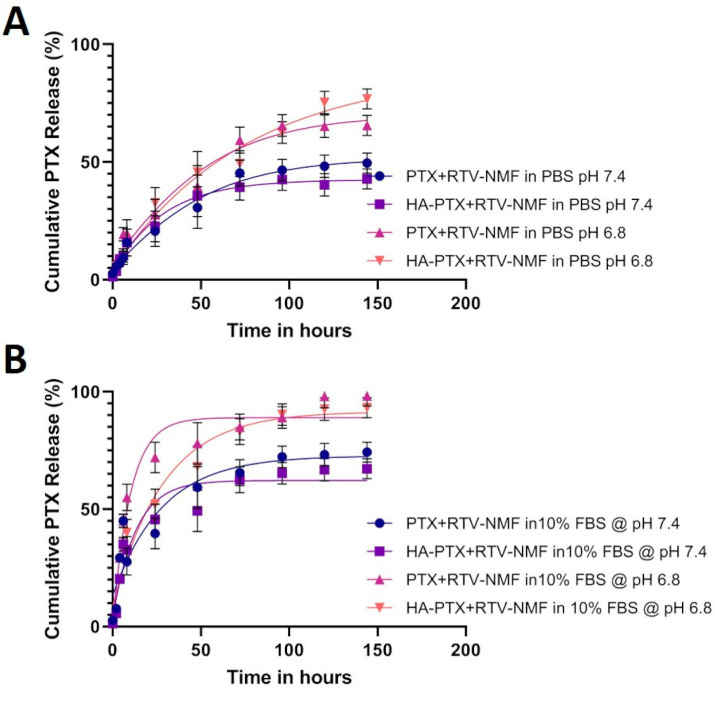
Cumulative PTX release from HA − PTX + RTV − NMF and PTX + RTV − NMF under various buffer conditions. (**A**) Cumulative PTX release from HA − PTX + RTV − NMF and PTX + RTV − NMF in PBB at pH 7.2 and pH 6.8, respectively, over a period of 6 days. (**B**) Cumulative PTX release from HA − PTX + RTV − NMF and PTX + RTV − NMF in 10% FBS solution in DPBB at pH 7.2 and pH 6.8, respectively, over a period of 6 days. The data were expressed as mean ± SD (*n* = 3).

**Figure 7 ijms-22-01257-f007:**
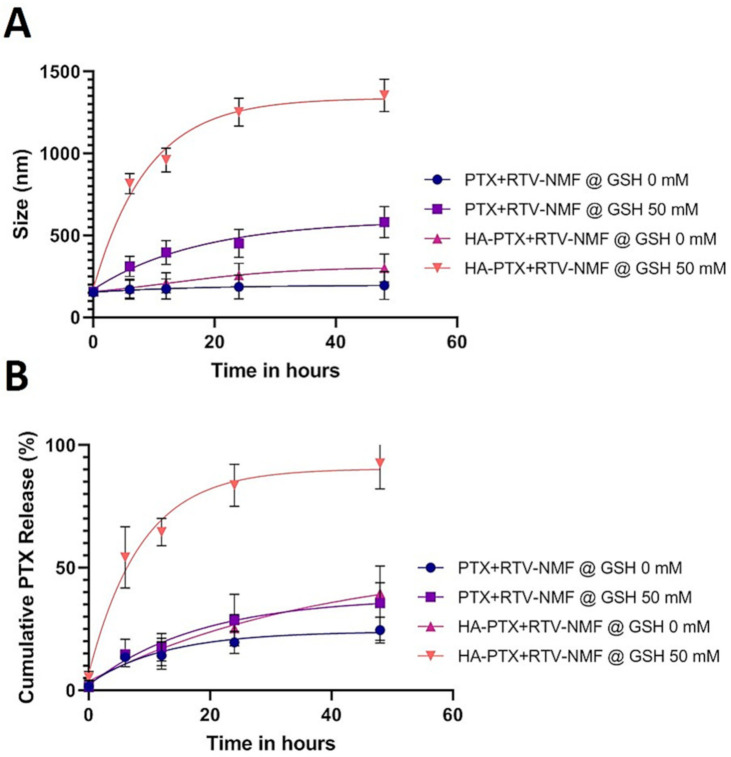
Nanomicelles in reduction stimulated environment under various reductive buffer conditions containing glutathione (GSH). (**A**) Determination of increase in size of HA − PTX + RTV − NMF and PTX + RTV − NMF in 50 Mm GSH in 1X PBS over 48 h. (**B**) Cumulative PTX release from HA − PTX + RTV − NMF and PTX + RTV − NMF in 50 Mm GSH in 1X PBS over 48 h. The data were expressed as mean ± SD (*n* = 3).

**Figure 8 ijms-22-01257-f008:**
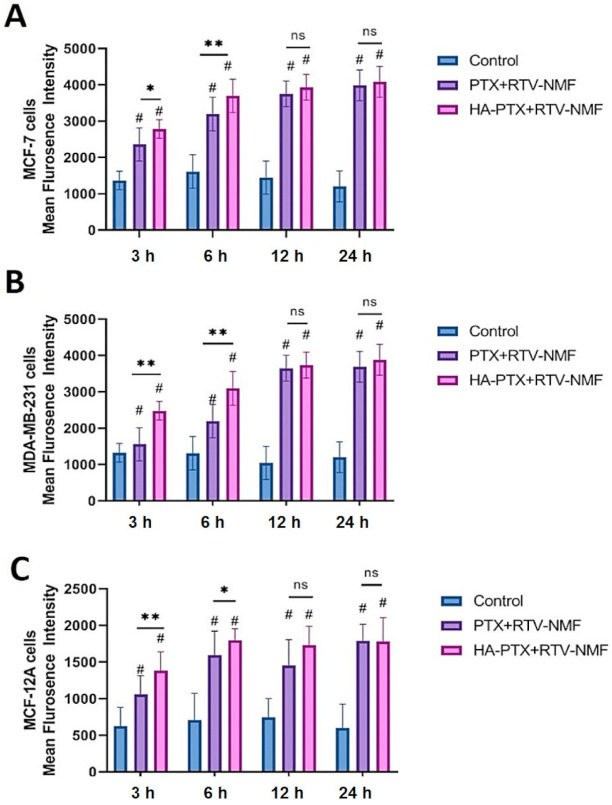
Time-dependent uptake of FITC-labeled HA − PTX + RTV − NMF solution and FITC-labeled PTX + RTV − NMF solution in breast cancer MCF-7 and triple-negative breast cancer, MDA-MB-231 cells. (**A**) Uptake of FITC-labeled HA − PTX + RTV − NMF solution and FITC-labeled PTX + RTV − NMF solution at 3, 6, 12, and 24 h in MCF-7 cells, (**B**) uptake of FITC-labeled HA − PTX + RTV − NMF solution and FITC-labeled PTX + RTV − NMF solution at 3, 6, 12, and 24 h in MDA-MB-231 cells. (**C**) Uptake of FITC-labeled HA − PTX + RTV − NMF solution and FITC-labeled PTX + RTV − NMF solution at 3, 6, 12, and 24 h in MCF-12A cells. The data were expressed as mean ± SD (*n* = 3). (# = significant difference as compared to control group and * *p* ≤ 0.05, ** *p* ≤ 0.01, ns = not significant as compared to PTX + RTV − NMF group).

**Figure 9 ijms-22-01257-f009:**
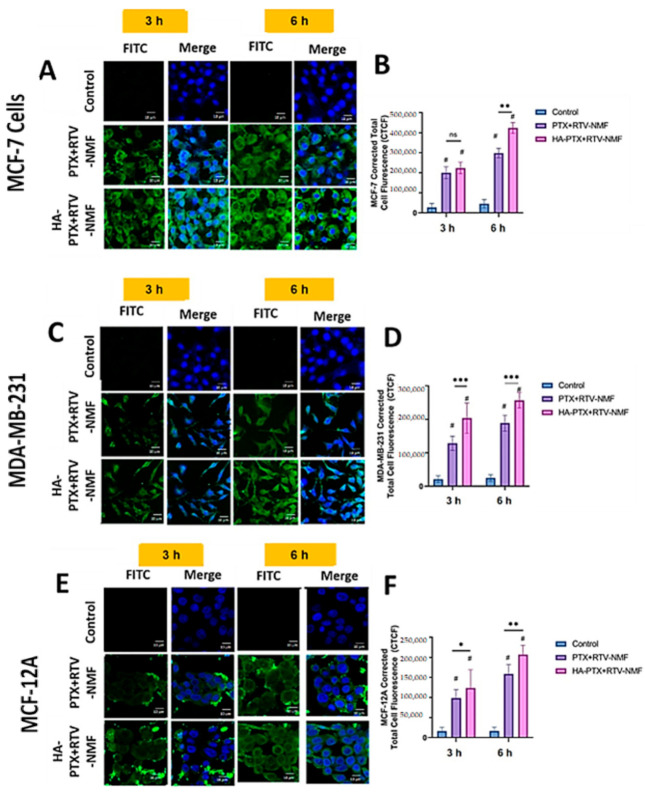
Time-dependent uptake of FITC-labeled HA − PTX + RTV − NMF and FITC-labeled PTX + RTV − NM Fin breast cancer MCF-7 and triple-negative breast cancer, MDA-MB-231, and normal breast epithelium MCF-12A cells. (**A**) Uptake of FITC-labeled HA − PTX + RTV − NMF solution and FITC-labeled PTX + RTV − NMF solution at 6 and 12 h in MCF-7 cells. (**B**) Corrected total cell fluorescence (CTCF) of FITC-labeled HA − PTX + RTV − NMF solution and FITC-labeled PTX + RTV − NMF solution at 6 and 12 h in MCF-7 cells. (**C**) Intracellular distribution of FITC-labeled HA − PTX + RTV − NMF and FITC-labeled PTX + RTV − NMF solution at 6 and 12 h. (**D**) CTCF of FITC-labeled HA − PTX + RTV − NMF solution and FITC-labeled PTX + RTV − NMF solution at 3 and 6 h. (**E**) Intracellular distribution of FITC-labeled HA − PTX + RTV − NMF and FITC-labeled PTX + RTV − NMF solution at 3 and 6 h. (**F**) CTCF of FITC-labeled HA − PTX + RTV − NMF solution and FITC-labeled PTX + RTV − NMF solution at 3 and 6 h. The data were expressed as mean ± SD (*n* = 3). (# = significant difference as compared to control group and * *p* ≤ 0.05, ** *p* ≤ 0.01, ns = not significant, *** *p* ≤ 0.001 as compared to PTX + RTV − NMF group.)

**Figure 10 ijms-22-01257-f010:**
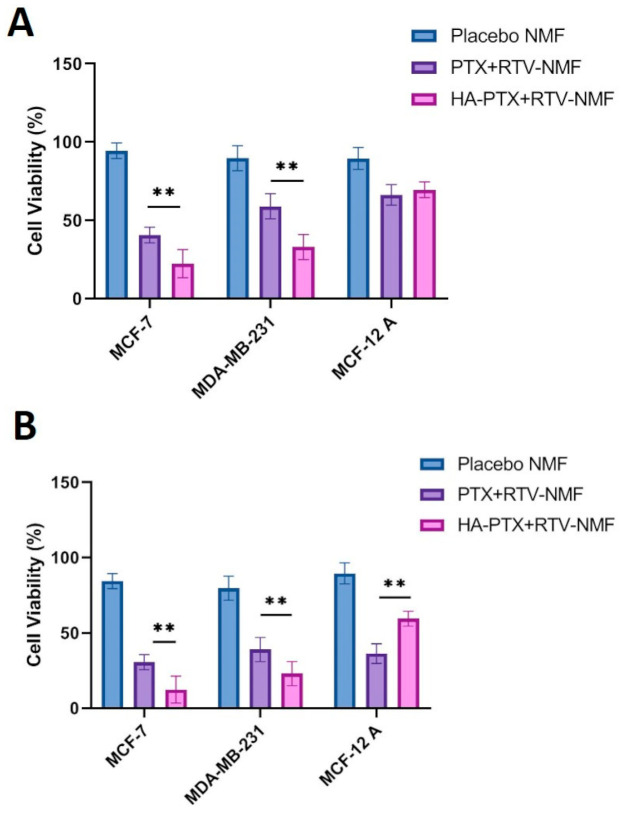
In vitro cell viability for nanomicellar formulations. (**A**) Cell viability (%) in MCF-7, MDA-MB-231, and MCF-12A cells treated with placebo NMF, PTX + RTV − NMF, and HA − PTX + RTV − NMF for 24 h. (**B**) Cell viability (%) in MCF-7, MDA-MB-231, and MCF-12A cells treated with placebo NMF, PTX + RTV − NMF, and HA − PTX + RTV − NMF for 72 h. The data were expressed as mean ± SD (*n* = 3) (** *p* ≤ 0.01, ns = not significant as compared to PTX + RTV − NMF group).

**Figure 11 ijms-22-01257-f011:**
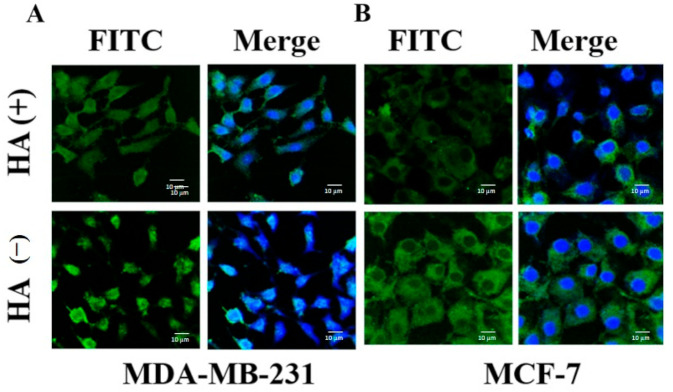
Confocal laser scanning microscopy (CLSM) images of (**A**) MDA-MB-231 cells incubated with FITC-labeled HA − PTX + RTV − NMF in the presence (+) and absence (−) of excess HA and (**B**) MCF-7 cells incubated with FITC-labeled HA − PTX + RTV − NMF in the presence (+) and absence (−) of excess HA.

**Figure 12 ijms-22-01257-f012:**
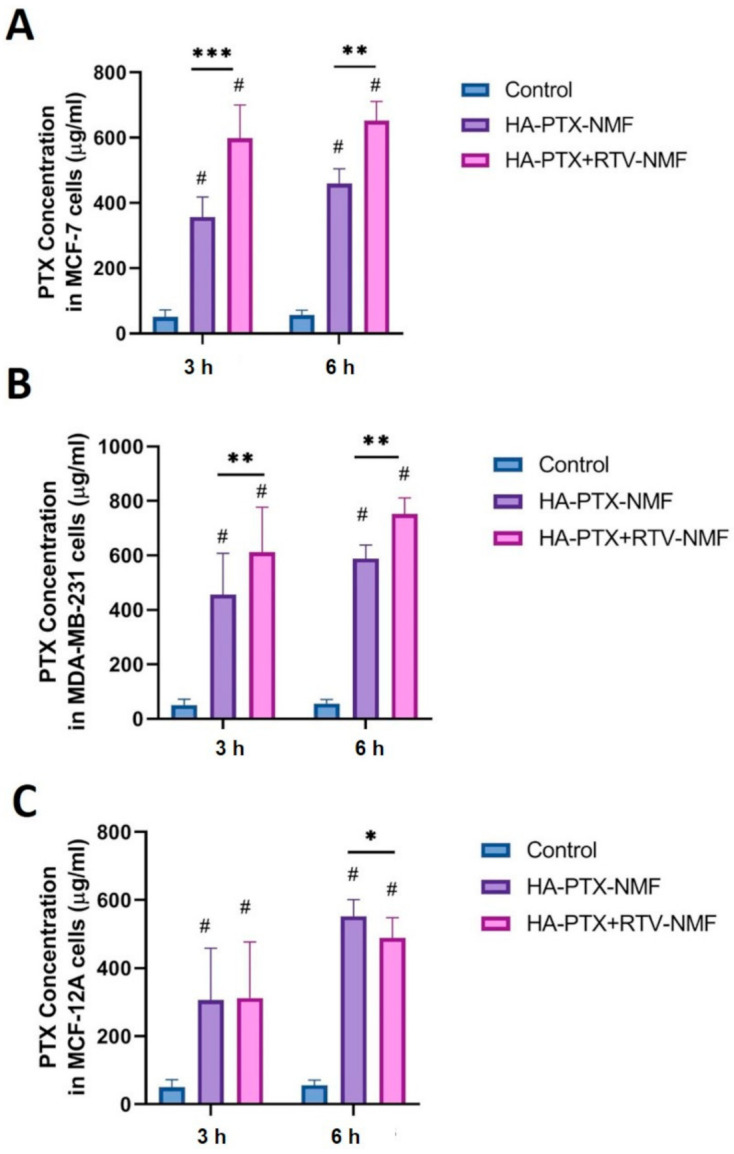
Intracellular accumulation of PTX in (**A**) MCF-7 and (**B**) MDA-MB-231 (**C**) MCF-12A cells at 6 and 12 h after treatment with HA − PTX + RTV − NMF and HA − PTX − NMF at 6 and 12 h using UHPLC-MS method of PTX quantification. The data were expressed as mean ± SD (*n* = 3). (# = significant difference as compared to control group and * *p* ≤ 0.05, ** *p* ≤ 0.01, ns = not significant, *** *p* ≤ 0.001 as compared to PTX + RTV − NMF group).

**Figure 13 ijms-22-01257-f013:**
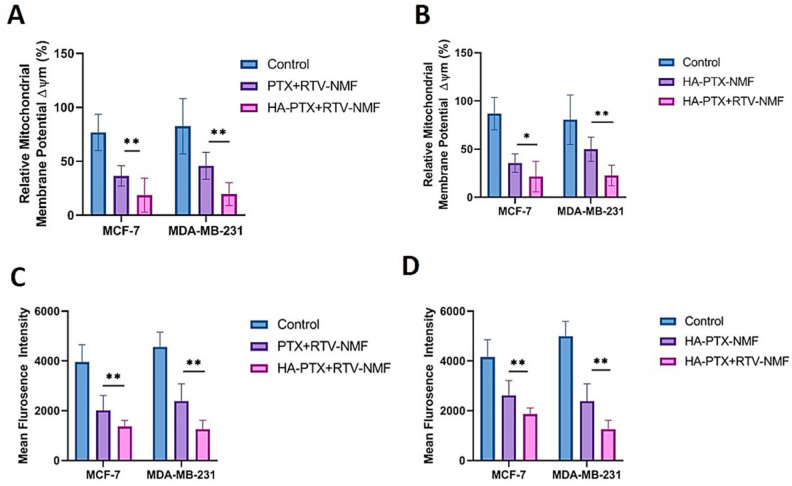
In vivo potency determination of HA − PTX + RTV − NMF. (**A**) Difference in mitochondrial membrane potential ΔΨm (JC-1 Red/Green ratio) in MCF-7 and MDA-MB-231 breast cancer cell lines treated with nanomicellar formulations like (i) HA − PTX + RTV − NMF and (ii) PTX + RTV − NMF treated for 24 h. (**B**) Difference in mitochondrial membrane potential ΔΨm (JC-1 Red/Green ratio) in MCF-7 and MDA-MB-231 breast cancer cell lines treated with nanomicellar formulations (i) HA − PTX + RTV − NMF and (ii) HA − PTX − NMF treated for 24 h. (**C**) Determination of reactive oxygen species (ROS) in MCF-7 and MDA-MB-231 breast cancer cell lines treated with (i) HA − PTX + RTV − NMF and (ii) PTX + RTV − NMF for 24 h. (**D**) Determination of ROS in MCF-7 and MDA-MB-231 breast cancer cell lines treated with (i) HA − PTX + RTV − NMF and (ii) HA − PTX − NMF for 24 h. The data were expressed as mean ± SD (*n* = 3). (* *p* ≤ 0.05 and ** *p* ≤ 0.01 as compared to PTX + RTV − NMF/ HA − PTX − NMF group).

**Table 1 ijms-22-01257-t001:** Full factorial design of experiment for hyaluronic acid-targeted, paclitaxel and ritonavir co-loaded nanomicellar formulation (HA − PTX+RTVNMF) using JMP^®^ software.

Formulation Name	Coded Design			Uncoded Design		
	X1	X2	X3	X1 = Sonication Time (min)	X2 = HA-ss-PLGA (wt%)	X3 = Vit E-TPGS (wt%)
F1	0	0	0	22.5	2	4
F2	0	0	0	22.5	2	4
F3	−	−	+	20	0.5	5
F4	−	+	+	20	3.5	5
F5	0	0	0	22.5	2	4
F6	+	−	+	25	0.5	5
F7	+	−	−	25	0.5	3
F8	−	−	−	20	0.5	3
F9	+	+	+	25	3.5	5
F10	−	+	−	20	3.5	3
F11	+	+	−	25	3.5	3

**Table 2 ijms-22-01257-t002:** Full factorial design of experiment using JMP software.

Formulation	Pattern	Sonication Time (min)	HA−PLGA (wt%)	Vit E TPGS(wt%)	Size (nm)	PDI	Zeta (mV)	PTX		RTV	
								EE%	LE%	EE%	LE%
F1	(0)(0)(0)	22.5	2	4	144.0	0.2	0.2	96.5	7.4	95.5	1.9
F2	(0)(0)(0)	22.5	2	4	146.0	0.2	0.2	95.7	7.2	96.9	1.8
F3	(−)(−)(+)	20	0.5	5	168.8	0.5	0.0	83.0	6.9	84.2	1.9
F4	(−)(+)(+)	20	3.5	5	256.2	0.6	0.0	88.9	4.9	90.0	1.3
F5	(0)(0)(0)	22.5	2	4	142.5	0.2	0.1	95.4	7.2	96.0	1.8
F6	(+)(−)(+)	25	0.5	5	159.6	0.8	0.2	85.8	7.1	86.7	1.6
F7	(+)(−)(−)	25	0.5	3	204.3	0.3	0.0	73.7	9.2	93.8	3.2
F8	(−)(−)(−)	20	0.5	3	179.7	0.3	0.0	74.0	9.3	93.4	3.2
F9	(+)(+)(+)	25	3.5	5	189.9	0.6	0.0	88.4	5.0	89.2	1.3
F10	(−)(+)(−)	20	3.5	3	249.8	0.5	0.2	76.9	5.5	78.7	1.5
F11	(+)(+)(−)	20	3.5	3	225.0	0.4	0.2	75.5	5.4	75.9	1.4

**Table 3 ijms-22-01257-t003:** HA − PTX+RTVNMF dilution study when stored at temperatures of 4 °C, room temperature (RT), and 37 °C for 24 h.

Dilution Factor	Hydrodynamic Size (nm)			Polydispersity Index (PDI)			Zeta Potential (mV)		
	4 °C	R.T.	37 °C	4 °C	R.T.	37 °C	4 °C	R.T.	37 °C
0	144.1	144.2	156.3	0.25	0.183	0.26	0.46	0.522	0.53
10	144.6	144.9	159.6	0.21	0.251	0.29	0.56	0.651	0.95
50	144.8	145.2	165.3	0.26	0.259	0.21	0.85	1.016	1.63
100	149.6	147.5	169.7	0.25	0.322	0.31	0.89	1.13	1.46
150	152.0	150.3	170.3	0.29	0.468	0.34	1.26	1.523	1.62
200	154.9	155.4	176.4	0.35	0.672	0.5	1.65	1.92	2.30

**Table 4 ijms-22-01257-t004:** HA − PTX + RTV − NMF -accelerated temperature stability study.

Time Points	Hydrodynamic Size (nm)			Polydispersity Index (PDI)			Zeta Potential (mV)		
	4 °C	25 °C	40 °C	4 °C	25 °C	40 °C	4 °C	25 °C	40 °C
Time 0	145.1	144.2	151.3	0.13	0.16	0.29	0.96	0.722	1.53
Day 3	149.6	165.9	397.6	0.22	0.25	0.56	1.6	1.651	4.95
Day 7	156.8	175.4	572.3	0.26	0.39	0.61	1.25	2.160	5.63

**Table 5 ijms-22-01257-t005:** HA − PTX + RTV − NMF-accelerated serum stability study at 37 °C.

Time Points	Hydrodynamic Size (nm)	Polydispersity Index (PDI)	Zeta Potential (mV)
Time 0	149.5	0.26	8.96
Day 3	163.6	0.45	12.3
Day 7	262.3	0.51	16.3

## Data Availability

Not applicable.

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
