# Peer review of "Hyaluronic Acid-Targeted Stimuli-Sensitive Nanomicelles Co-Encapsulating Paclitaxel and Ritonavir to Overcome Multi-Drug Resistance in Metastatic Breast Cancer and Triple-Negative Breast Cancer Cells"

_ijms, 2021, doi:10.3390/ijms22031257_

Round 1

Reviewer 1 Report

The manuscript of Vrinda Gote et.al. presents a new technology forproducing "smart" hyaluronic acid (HA) decorated mixed nanomicelles encapsulating chemotherapeutic agent paclitaxel (PTX) and P-glycoprotein inhibitor ritonavir (RTV) which target CD44 receptor overexpressed on breast cancer cells.  The final aim is to enhance the target-delivery of the chemotherapeutic agent paclitaxel while decreasing the multi-drug resistance characterizing the highly metastatic breast cancer (MBC) and triple negative breast cancer (TNBC).

Overall the paper is very well written with details on the organic chemistry for nanomicelles preparation together with accompanying physical chemical methods used to characterize the size, morphology, zeta potential, drug entrapment and drug release properties of the designed nanomicelles. In addition some cellular assays are presenting to show the ability of these smart designed nanomicelles to induce intracellular ROS, mitochondrial stress which in turn can contribute to the additional anti-proliferative effects mediated by paclitaxel.

The authors could have used additional assay to investigate the effect of the designed nanomicelles on the CD44-mediated signal transduction including  PKN2, the Rho-GTPases RAC1 and RHOA, Rho-kinases and phospholipase C known to coordinate signaling pathways promoting calcium mobilization and actin-mediated cytoskeleton reorganization essential for cell migration and adhesion. Additionally, the authors could improve the read-out for their designed nanomicelles by looking on other side effects of paclitaxel mediated inhibition of cell proliferation: such as confocal microscopy analysis of microtubule assembly and cell cytokinesis, etc.

These additional analyses could be the subject of a future research which will enhance the molecular mechanism underlying the action of these nanomicelles.

Altogether, the presented biochemical, cellular, biophysical and physical chemical assays are very well documented with appropriate statistical and bioinformatics analysis and thus the current presentation of this manuscript is suitable for being accepted for publication in the IJMS journal.

Author Response

The authors highly appreciate the insightful comments and thorough review by the reviewer. The future studies mentioned by the reviewer are very interesting and have given the authors directions for further analysis of this nanomicellar formulation. Such insightful molecular biology studies have immensely increased the author's knowledge about how such studies can be utilized as a tool for further evaluation not only for this formulation but can be useful for other studies. 

 Figure 10 is updated (Ctrl is changed to Control). Minor changes in grammatical and spellings errors are corrected in the revised manuscript. 

The authors appreciate the careful review of all biochemical, cellular, biophysical and physical-chemical assays presented in this manuscript. 

Reviewer 2 Report

Presented work represent strongly systematic and purposeful approach to interesting problematic. Therefore, I think that this work is suitable for your journal. Nevertheless, few points can be done for the increase its impact.

  • Authors construct model for the prediction of nanoparticles properties. Will this model validate based on the experimental data? It was possible show comparison between theoretical and experimental data?
  • In Figure 5 and 6 are showed experimental data and corresponding calculated curves. Based on what model were these calculated? It possible this curves calculated by used theoretical model?

Minor

Lines 36, 54, 68, 75, 106, 115, 130, 218, 284, 333, 336, 370, 377, 415, 550, 594, 665,674, 677, 682, 737, 766. 935- double space

Lines 846-850 error conversion?

Line 59 can be prevented..

Author Response

  • The authors highly appreciate the insightful comments of the reviewer. This will definitely help in improving the quality of the manuscript. 
  •  
  • Authors construct model for the prediction of nanoparticles properties. Will this model validate based on the experimental data? It was possible show comparison between theoretical and experimental data?
  • Answer : design of experiment - full factorial design by JMP software was used to predict which formulation has the set optimized parameters. This model was evaluated by experimentation and statistical model. Specifically here, the prediction profiler was plotted. This indicated how well the experimenal findings are in accordance with the mathematical model outcomes. This is now explained ed in section 2.2 in the revised manuscript. 

  • In Figure 5 and 6 are showed experimental data and corresponding calculated curves. Based on what model were these calculated? It possible this curves calculated by used theoretical model.
  • Answer : No mathematical model was used here to represent data from Figure 6 and 5. Here line curves were plotted for PTX released in the external buffer solution in the case in Figures 5 and 6. Figure 5 indicates what will be the in vitro release of the formulation under various buffers at various solution pH. While in Figure 6, PTX release from the nanomicelles was calculated in an external reductive environment of glutathione (GSH). 
  •  
  • Lines 36, 54, 68, 75, 106, 115, 130, 218, 284, 333, 336, 370, 377, 415, 550, 594, 665,674, 677, 682, 737, 766. 935- double space 
  • This is now corrected. 

Lines 846-850 error conversion?

This is now corrected in the revised manuscript. 

Line 59 can be prevented..

This is now corrected in the revised manuscript.